# Graded Ca²⁺/calmodulin-dependent coupling of voltage-gated CaV1.2 channels

Rose E Dixon[1], Claudia M Moreno[1], Can Yuan[1], Ximena Opitz-Araya[1], Marc D Binder[1], Manuel F Navedo[2]*, Luis F Santana[1]*

[1]Department of Physiology and Biophysics, University of Washington, Seattle, United States; [2]Department of Pharmacology, University of California, Davis, Davis, United States

**Abstract** In the heart, reliable activation of Ca²⁺ release from the sarcoplasmic reticulum during the plateau of the ventricular action potential requires synchronous opening of multiple CaV1.2 channels. Yet the mechanisms that coordinate this simultaneous opening during every heartbeat are unclear. Here, we demonstrate that CaV1.2 channels form clusters that undergo dynamic, reciprocal, allosteric interactions. This 'functional coupling' facilitates Ca²⁺ influx by increasing activation of adjoined channels and occurs through C-terminal-to-C-terminal interactions. These interactions are initiated by binding of incoming Ca²⁺ to calmodulin (CaM) and proceed through Ca²⁺/CaM binding to the CaV1.2 pre-IQ domain. Coupling fades as $[Ca^{2+}]_i$ decreases, but persists longer than the current that evoked it, providing evidence for 'molecular memory'. Our findings suggest a model for CaV1.2 channel gating and Ca²⁺-influx amplification that unifies diverse observations about Ca²⁺ signaling in the heart, and challenges the long-held view that voltage-gated channels open and close independently.

*For correspondence:
mfnavedo@ucdavis.edu (MFN);
santana@uw.edu (LFS)

**Competing interests:** The authors declare that no competing interests exist.

## Introduction

L-type Ca²⁺ channels are composed of a pore-forming $\alpha_1$ subunit and four additional accessory subunits ($\alpha_2$, $\beta$, $\gamma$, $\delta$) (*Catterall, 1995*). Four different $\alpha_1$ subunits have been identified to date, one of which, CaV1.2, is expressed in neurons as well as cardiac and arterial smooth muscle (*Koch et al., 1990*; *Navedo et al., 2007*; *Zhang et al., 2007*). Four distinct genes encode L-type Ca²⁺ channel $\beta$-subunits, each with multiple splice variants. In addition, four $\alpha_2\delta$ genes have been identified. Both cell- and tissue-specific combinations of these CaV1.2 subunits endow the channels with distinct functional properties (*Catterall, 2000*).

A prominent characteristic of CaV1.2 channels is the tight regulation of their activity by the Ca²⁺ signals they produce (*Ben-Johny and Yue, 2014*). For example, increases in $[Ca^{2+}]_i$ have been implicated in CaV1.2 facilitation; this Ca²⁺-dependent facilitation (CDF) is a form of positive feedback that amplifies Ca²⁺ influx. An increase in intracellular Ca²⁺ concentration ($[Ca^{2+}]_i$) has also been proposed to exert the opposite effect—Ca²⁺-dependent inactivation (CDI). Thus, the balance between CDF and CDI of CaV1.2 channels plays a key role in regulating the magnitude of Ca²⁺ influx. The general consensus is that CDF and CDI involve Ca²⁺ binding to calmodulin (CaM) in the IQ domain in the C-terminal tail of these channels.

During excitation-contraction (EC) coupling, membrane depolarization opens CaV1.2 channels in the sarcolemma of ventricular myocytes. This allows a small amount of Ca²⁺ to enter ventricular myocytes that can be detected optically in the form of a 'CaV1.2 sparklet', raising local $[Ca^{2+}]_i$ beyond the threshold for activation of ryanodine receptors (RyRs) in the sarcoplasmic reticulum (*Wang et al., 2001*). Synchronous activation of multiple RyRs by CaV1.2 channels produces a global rise in $[Ca^{2+}]_i$ that initiates myocardial contraction (*Cheng et al., 1993*).

**eLife digest** To pump blood around the body, the muscle cells within the heart must contract and relax together with a regular rhythm. A contraction begins when proteins called $Ca_V1.2$ channels embedded in the cell membranes of heart cells open to allow calcium ions to enter the cells. The calcium ions that enter through these $Ca_V1.2$ channels trigger the release of calcium ions from storage compartments within the cells, which leads to the heart contracting. However, to trigger this release of calcium ions, many $Ca_V1.2$ channels have to open at the same time and we do not yet know how this is co-ordinated.

Dixon et al. studied $Ca_V1.2$ channels in heart muscle cells from mice. The experiments show that these proteins are arranged in clusters of eight, on average, in the cell membrane. When calcium ions enter the cell they bind to a protein called calmodulin, which in turn binds to a $Ca_V1.2$ channel. This allows the $Ca_V1.2$ channels within a cluster to interact with each other. The physical association between $Ca_V1.2$ channels within clusters enables them to work cooperatively; they open at the same time to allow more calcium ions to enter and then close together to allow the cell to relax.

Dixon et al. found that even when levels of calcium ions in the cells declined, the $Ca_V1.2$ channels within clusters remained open for a little while longer before they closed. This suggests that the interactions between the $Ca_V1.2$ channels act as a type of 'molecular memory' that may alter how the cells respond to future activity.

These results challenge the previously held view that the $Ca_V1.2$ channels open and close independently of one another. Future studies will seek to understand the molecular details of how these channels cluster together, and how this clustering affects changes in heart rate and heart abnormalities like long QT syndrome.

EC coupling in ventricular myocytes is remarkably reproducible, with each action potential (AP) invariably evoking a whole-cell $[Ca^{2+}]_i$ transient that results in contraction. At the membrane potentials reached during the plateau of the ventricular AP (approximately +50 mV), the driving force for $Ca^{2+}$ entry at physiological $Ca^{2+}$ levels (~2 mM) is so low that opening of a single $Ca_V1.2$ channel is not sufficient to raise local $[Ca^{2+}]_i$ beyond the RyR activation threshold. However, the probability of RyR activation during this phase of the AP is very high (>0.9). This degree of reliability would presumably require 5–10 $Ca_V1.2$ channels to open simultaneously (*Inoue and Bridge, 2003*; *Sobie and Ramay, 2009*). However, because the maximum open probability ($P_o$) of $Ca_V1.2$ channels at physiological $[Ca^{2+}]_o$ is ~0.3 (*Josephson et al., 2010*), the probability of 5–10 independently gating channels opening simultaneously is extremely low (i.e., $0.3^5$ to $0.3^{10}$). This raises a fundamental question: if the probability of coincident openings of the requisite number of $Ca_V1.2$ channels is so low, why is the probability of RyR activation during the cardiac AP so high? Answering this question is critical for understanding the mechanistic basis of reliable cardiac performance.

A potential answer to this conundrum lies in the recently proposed concept that clusters of $Ca_V1.2$ channels can be functionally coupled to one another through physical interactions between their C-terminal tails (*Dixon et al., 2012*). This interaction enables physically linked channels to coordinate their gating, leading to amplification of $Ca^{2+}$ influx, $Ca^{2+}$ current facilitation, and EC coupling in ventricular myocytes. Importantly, this model challenges the long-held assumption that $Ca_V1.2$ channels, like other classes of voltage-gated channels, function exclusively as monomers that gate independently of one another. To date, however, the mechanism underlying functional $Ca_V1.2$ channel coupling has remained unknown.

Here, using a combination of super-resolution nanoscopy, $Ca^{2+}$-imaging, electrophysiology and two methodologically distinct assays of protein–protein interaction, we have assembled a body of evidence that significantly alters our current understanding of $Ca^{2+}/CaM$ regulation of $Ca_V1.2$ channels and reconciles diverse observations about $Ca^{2+}$ signaling in ventricular myocytes. We discovered that binding of $Ca^{2+}$ to CaM induces C-terminal-to-C-terminal $Ca_V1.2$ channel interactions that increase the activity of adjoined channels, facilitating $Ca^{2+}$ currents and increasing $Ca^{2+}$ influx. Notably, we found that functional coupling outlasts the $Ca^{2+}$ current that evokes it, providing evidence for a type of 'molecular memory' that could transiently shape the response of the cell to subsequent APs. We propose that cooperative gating of $Ca_V1.2$ channels is

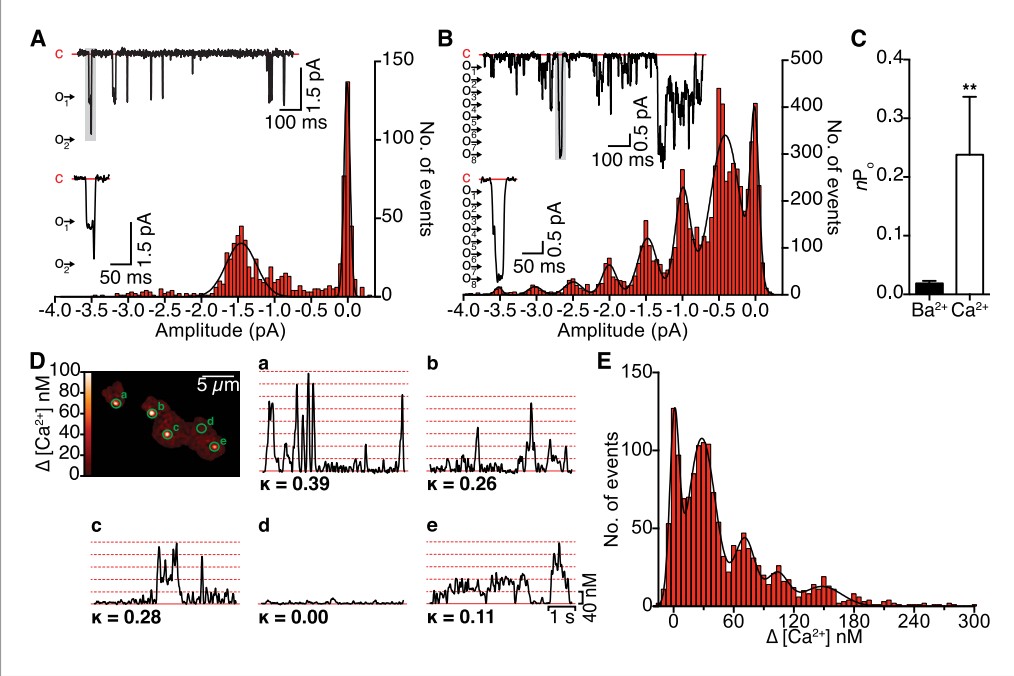

**Figure 1**. Single-channel electrical and optical recordings of $Ca_V1.2$ channel coupling. (**A** and **B**) Representative $i_{Ba}$ (**A**) and $i_{Ca}$ (**B**) traces recorded from $Ca_V1.2$-expressing tsA-201 cells during step depolarizations from −80 to −30 mV. Amplitude histograms (constructed from $n = 6$ cells each) were fit with multi-component Gaussian functions (solid black lines). A portion of each trace (gray box) is shown enlarged below, showing that the resulting L-type $Ca^{2+}$ current reflects the simultaneous opening and closing of multiple channels with $Ca^{2+}$ as the charge carrier, but not with $Ba^{2+}$ as the charge carrier. (**C**) Bar chart of $i_{Ba}$ and $i_{Ca}$ single-channel activity ($nP_o$). Data are presented as means ± SEM (**p < 0.01). (**D**) Calibrated TIRF image of an adult ventricular myocyte dialyzed with the $Ca^{2+}$ indicator dye Rhod-2 via the patch pipette (see also *Video 1*). Time courses of $[Ca^{2+}]_i$ from each sparklet site (indicated by green circles on TIRF image) and their κ values are shown in panels **a**-**e**. (**E**) All-points histogram of $Ca^{2+}$ sparklet data recorded from adult ventricular myocytes. The data were fit with a multi-component Gaussian function (solid black line).

a new general mechanism for the regulation of excitability and $Ca^{2+}$ influx in cardiac myocytes and suggest that this concept can be extended to other excitable cells.

## Results

### $Ca^{2+}$ ions augment functional coupling of $Ca_V1.2$ channels

We began our study by expressing $Ca_V1.2$ channels in tsA-201 cells and recording elementary $Ca_V1.2$ currents from cell-attached patches. Currents were elicited with a step depolarization to −30 mV with $Ba^{2+}$ or $Ca^{2+}$ as the charge carrier (*Figure 1A,B*). The mean amplitudes of $i_{Ca}$ and $i_{Ba}$ were 0.50 ± 0.02 ($n = 6$) and 1.45 ± 0.01 pA ($n = 6$), respectively. All-points histograms revealed that multi-channel openings were more likely with $Ca^{2+}$ than with $Ba^{2+}$. Accordingly, the activity ($nP_o$), defined as the number of channels ($n$) times the open probability ($P_o$), of $Ca_V1.2$ channels within a patch, was significantly higher with $Ca^{2+}$ (0.24 ± 0.10) than $Ba^{2+}$ (0.02 ± 0.01; *Figure 1C*). Closer inspection of the multi-channel openings revealed that, with $Ca^{2+}$ as the charge carrier, multiple channels frequently opened together instantaneously and subsequently closed together. For example, in the enlarged trace in *Figure 1B*, four channels opened simultaneously, followed by the opening of four additional channels. The eight channels remained open for a time (~11.5 ms), then all closed simultaneously. This apparent coordinate gating of multiple $Ca_V1.2$ channels was not observed when $Ba^{2+}$ was used as the charge carrier. These results challenge the long-held and generally accepted view that individual $Ca_V1.2$ channels gate independently, and instead strongly suggest that these channels frequently exhibit 'cooperative gating'. An additional implication is that $Ca^{2+}$ itself increases the probability of cooperative $Ca_V1.2$ channel gating.

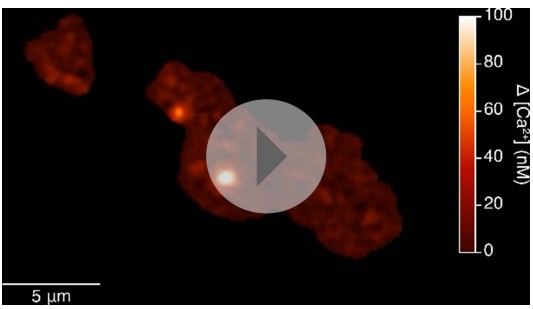

**Video 1.** Cardiomyocyte Ca²⁺ sparklets. Stack of 2D images acquired at 100 Hz from a whole-cell patch-clamped adult ventricular myocyte held at −80 mV and dialyzed with Rhod-2 via the patch pipette.

To test whether this hypothesis holds true in primary cells, we recorded Ca$_V$1.2 Ca²⁺ sparklet activity in freshly isolated adult ventricular myocytes (*Figure 1D* and *Video 1*). A multi-Gaussian fit to the all-points histogram of the calibrated Ca²⁺ signal obtained with 20 mM [Ca²⁺]$_o$ revealed quantal amplitudes of 36.8 ± 4.2 nM (*Figure 1E*), in excellent agreement with the ~38 nM reported previously for Ca$_V$1.2 Ca²⁺ sparklets in arterial smooth muscle cells and tsA-201 cells (*Navedo et al., 2005*, *2006*). Consistent with our single-channel data, we frequently observed multi-quantal Ca²⁺ sparklets corresponding to the simultaneous opening of several Ca$_V$1.2 channels. We next applied a coupled Markov chain model to determine whether these Ca²⁺-influx events were solely attributable to stochastic, independently gating Ca$_V$1.2 channels or instead reflected cooperative channel gating. The model assigns a coupling coefficient (κ) value to each Ca²⁺ sparklet site, ranging from 0 for independently gating channels to 1 for channels that gate exclusively in a cooperative manner (*Chung and Kennedy, 1996*; *Navedo et al., 2010*). Using κ > 0.1 as a threshold for cooperative gating (*Navedo et al., 2010*), we found that the majority of Ca²⁺ sparklet sites displayed cooperative or 'coupled' gating behavior, consistent with our hypothesis. Together, these data suggest that functional coupling of Ca$_V$1.2 channels is a Ca²⁺-dependent phenomenon.

## Ca$_V$1.2 channels form clusters in ventricular myocytes

If the trigger for Ca$_V$1.2 channel coupling were a local elevation in [Ca²⁺]$_i$ resulting from the opening of a single channel, then the efficacy of this signal in recruiting a nearby channel would be critically dependent on the distance separating the channels. Using super-resolution nanoscopy (*Folling et al., 2008*), we examined the spatial organization of endogenous Ca$_V$1.2 channels in ventricular myocytes (*Figure 2A–C*) and heterologously expressed Ca$_V$1.2 channels in tsA-201 cells (*Figure 3A–C*). Our data clearly show that Ca$_V$1.2 channels formed clusters along the sarcolemmal Z-lines of ventricular myocytes (*Figure 2A,B*) and throughout the plasma membrane (PM) of tsA-201 cells (*Figure 3A,B*). The average area occupied by a Ca$_V$1.2 channel cluster was 2555 ± 82 nm² in ventricular myocytes (n = 5; *Figure 2C*) and 2190 ± 20 nm² in tsA-201 cells (n = 9; *Figure 3C*).

Since Ca$_V$1.2 channel clusters were localized to the t-tubule regions of ventricular myocytes where the junctional SR (jSR) comes into close apposition to the myocyte PM, one might predict that the channel clusters would similarly localize to PM-adjacent ER structures in tsA-201 cells. To test this idea, we co-expressed mCherry-sec61β (a general ER marker (*Zurek et al., 2011*)) in tsA-201 cells together with Ca$_V$1.2 channels. Super-resolution imaging revealed that Ca$_V$1.2 channel cluster distribution was not restricted to regions of the surface membrane where the ER was located (*Figure 3—figure supplement 1A*). Instead, these channels were broadly distributed along the PM surface. It is possible that the lack of organization of Ca$_V$1.2 channels along ER junctions in tsA-201 cells reflects a lack of contact points or excessive distance between the ER and the PM. In ventricular myocytes, the membrane binding protein junctophilin-2 (JPH2) has been suggested to tether the jSR to the t-tubule membrane (*Takeshima et al., 2000*). Thus, we attempted to anchor the ER to the PM by co-transfecting tsA-201 cells with JPH2. However, even in the presence of JPH2, the Ca$_V$1.2 channel cluster distribution was not limited to PM-ER junctions in the manner that they are in cardiomyocytes (*Figure 3—figure supplement 1B*). These data suggest that Ca$_V$1.2 channel clustering occurs independently of SR/ER microdomains.

To determine the number of channels within Ca$_V$1.2 clusters, we injected mice with an adeno-associated virus serotype 9 (AAV9) designed to express photo-activatable-GFP–tagged Ca$_V$β$_{2a}$, and examined ventricular myocyte Ca$_V$1.2 clusters 5 wk later using single-particle photobleaching (*Ulbrich and Isacoff, 2007*). Ca$_V$β$_{2a}$ is a palmitoylated peripheral membrane protein that binds to the α₁ pore-forming subunit of Ca$_V$1.2 with a 1:1 stoichiometry (*Dalton et al., 2005*); therefore,

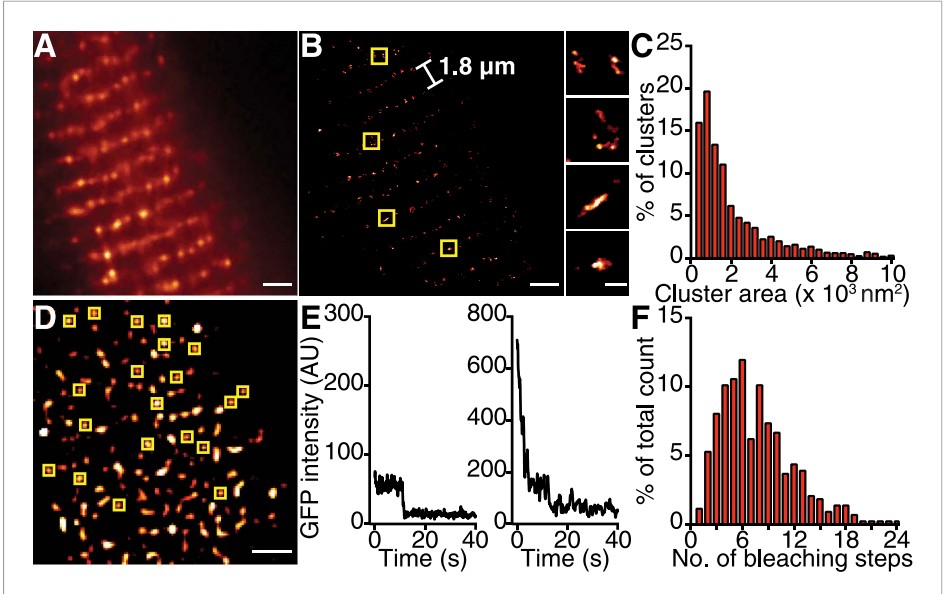

**Figure 2**. Ca_V1.2 channels form clusters in the ventricular myocyte PM. (**A**) TIRF image of a fixed, adult mouse ventricular myocyte immunolabeled with an antibody specific for Ca_V1.2 channels. (**B**) Super-resolution GSD image of the same cell. Channels are located along the t-tubule network with the characteristic 1.8-μm separation. Yellow boxes denote location of higher-magnification images of channel clusters (*right*). (**C**) Distribution of cluster areas in ventricular myocytes (*n* = 5 myocytes). (**D**) An average of the first five frames of a TIRF image time series taken of a myocyte isolated from mice expressing Ca_Vβ_2a-PA-GFP. Yellow boxes indicate spots selected for analysis. Scale bars = 2 μm. (**E**) Examples of bleaching steps for Ca_Vβ_2a-PA-GFP associated with Ca_V1.2 channels. (**F**) Distribution of bleaching steps obtained from 435 spots selected from *n* = 11 cells.

photo-activation of this protein with 405-nm light provides a fluorescent marker of Ca_V1.2 channels. Single Ca_V1.2 were identified and excited using total internal reflection fluorescence (TIRF) microscopy. The number of channels in each cluster was determined by continuous photobleaching and counting of stepwise decreases in fluorescence intensity (*Figure 2D–F*). A preponderance (47%) of Ca_V1.2 clusters displayed 1 to 6 stepwise decreases in fluorescence (*Figure 2F*). A single photobleaching step was observed in only 1% of the spots analyzed. Indeed, the mean number of bleaching steps per cluster was 7.91 ± 0.23 (*n* = 25). This suggests that Ca_V1.2 channels preferentially cluster in groups of about 8 channels in adult ventricular myocytes. We did not discriminate cluster size based on location; thus, our estimate of 8 channels/cluster combines dyadic and extra-dyadic populations.

Similar stepwise photobleaching experiments were performed on enhanced green fluorescent protein (EGFP)-tagged Ca_V1.2 channels heterologously expressed in tsA-201 cells (*Figure 3D–F*). TIRF imaging showed that Ca_V1.2 clusters displayed a mean of 5.07 ± 0.15 (*n* = 10) discrete bleaching steps. Taken together with the data from ventricular myocytes, these findings suggest that the formation of multi-channel clusters is a fundamental property of Ca_V1.2 channels with important implications for Ca^{2+} signaling.

## Spontaneous Ca_V1.2 channel coupling occurs via Ca^{2+}-dependent physical interactions between adjacent channel C-termini

To investigate the mechanisms regulating Ca_V1.2-Ca_V1.2 interactions in living cells (*Figure 4*), we applied a bimolecular fluorescence complementation approach using Ca_V1.2 channels fused with either the N- or C-terminus of the split-Venus fluorescent protein system to yield Ca_V1.2-VN155(I152L) and Ca_V1.2-VC155, respectively. In isolation, VN155(I152L) and VC155 are non-fluorescent; however, when brought into close proximity by interacting proteins, they can reconstitute a full, fluorescent Venus protein. Thus, Venus fluorescence can be used to report spontaneous interactions between adjacent Ca_V1.2 channels, as depicted in *Figure 4A*. In cells expressing Ca_V1.2-VN155(I152L) and Ca_V1.2-VC155 channels, Venus fluorescence at −80 mV was very low, suggesting that Ca_V1.2-Ca_V1.2

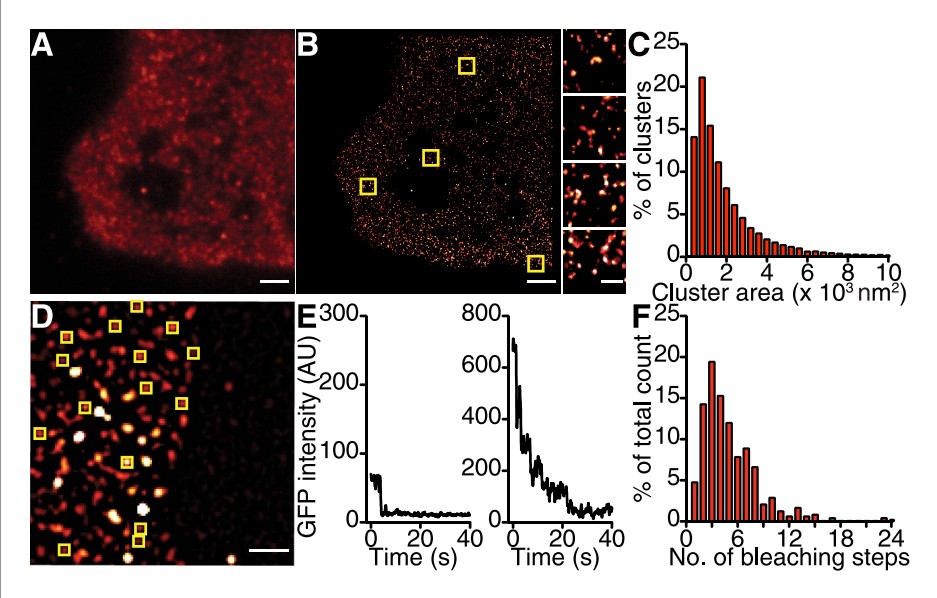

**Figure 3**. Ca$_V$1.2 channels form clusters in tsA-201 cell membranes. (**A** and **B**) TIRF and GSD images of immunolabeled Ca$_V$1.2 channels in a transfected tsA-201 cell (**A**). Yellow boxes in (**B**) indicate the location of each higher-magnification image (*right*). (**C**) Distribution of cluster areas in tsA-201 cells (*n* = 9 cells). (**D**) An average of the first five frames of a TIRF image time series for a tsA-201 cell expressing Ca$_V$1.2-EGFP. (**E**) Examples of bleaching steps for Ca$_V$1.2-EGFP. Scale bars = 2 μm. (**F**) Distribution of bleaching steps obtained from 484 spots selected from *n* = 10 cells.

The following figure supplement is available for figure 3:

**Figure supplement 1**. Ca$_V$1.2 channel clusters are not co-localized to tsA-201 ER structures.

channel interactions are rare at this hyperpolarized membrane potential. To determine whether an increase in [Ca$^{2+}$]$_i$ is *sufficient* to induce Ca$_V$1.2-Ca$_V$1.2 channel interactions, we loaded tsA-201 cells with DMNP-EDTA (caged Ca$^{2+}$) via the patch pipette and induced flash photolysis of DMNP-EDTA with pulses of 405-nm light while holding cells at −80 mV. Photolysis of DMNP-EDTA induced a transient increase in [Ca$^{2+}$]$_i$ and a concomitant increase in Venus fluorescence (*Figure 4—figure supplement 1*), demonstrating that an elevation in [Ca$^{2+}$]$_i$ is indeed *sufficient* to induce Ca$_V$1.2-Ca$_V$1.2 interactions.

To determine if Ca$^{2+}$ influx via Ca$_V$1.2 channels is *required* for channel interactions, we depolarized cells and recorded Venus fluorescence and membrane currents in the presence of Ba$^{2+}$ or Ca$^{2+}$. Cells were dialyzed with an intracellular solution containing 10 mM EGTA to restrict the local [Ca$^{2+}$]$_i$ signal to about 1 μm from the channel and maintain very low global [Ca$^{2+}$]$_i$. With Ba$^{2+}$ in the external solution, depolarization evoked currents (I$_{Ba}$) over a wide range of potentials, but Venus fluorescence was very low at all membrane potentials (*Figure 4B,F,G* and *Figure 4—figure supplement 2A–C*). After switching to a Ca$^{2+}$-containing external solution, application of the same voltage protocol activated currents (I$_{Ca}$) and induced graded increases in Venus fluorescence (*Figure 4C,F,G* and *Figure 4—figure supplement 2D–F*). The fluorescence-voltage and I$_{Ca}$ conductance-voltage relationships were sigmoidal. The normalized conductance and Venus fluorescence exhibited similar voltage dependencies (*Figure 4—figure supplement 2E,F*). Taken together with super-resolution, photobleaching and DMNP-EDTA data, these findings suggest that local and global [Ca$^{2+}$]$_i$ signals produced by Ca$^{2+}$ influx via Ca$_V$1.2 channels are *required* for physical interactions between adjacent channels within a cluster.

## Ca$^{2+}$/CaM binding to the pre-IQ domain mediates functional coupling, but not clustering, of Ca$_V$1.2 channels

We next investigated the mechanisms underlying Ca$^{2+}$-dependent coupling of Ca$_V$1.2 channels, focusing on CaM since this protein binds Ca$^{2+}$, associates with Ca$_V$1.2 C-terminal pre-IQ and IQ

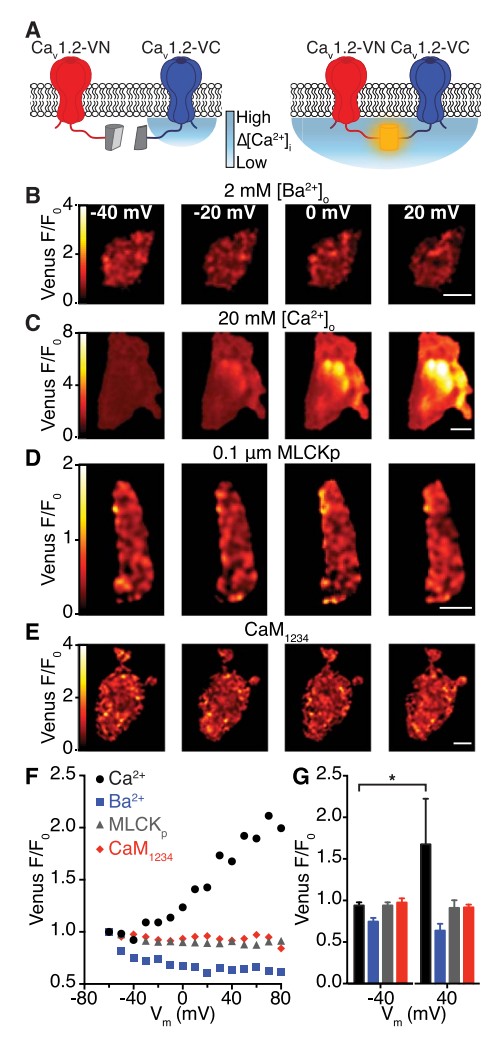

**Figure 4**. Interactions between Ca$_V$1.2 channel C-termini occur spontaneously and in a Ca$^{2+}$/CaM-dependent manner. (**A**) Illustration of the bimolecular fluorescence complementation strategy for assaying interactions between Ca$_V$1.2 channel C-termini. Non-interacting channels tagged at their C-terminus with either the N- or C-terminal half of split Venus are non-fluorescent (*left*). Spontaneous interactions between channel C-termini result in reconstitution of Venus and emission of fluorescence (*right*). (**B–E**) TIRF images obtained from whole-cell patch-clamped tsA-201 cells expressing Ca$_V$1.2-VN and Ca$_V$1.2-VC over 9-s voltage steps to the indicated potentials. Images were median-filtered, smoothed, pseudo-colored with a 'red-hot' LUT, and divided by the initial −60 mV image to obtain calibrated Venus F/F$_0$. Experiments were performed with 2 mM Ba$^{2+}$ (**B**) or 20 mM Ca$^{2+}$ (**C–E**) in the perfusing solution. Scale bars = 3 µm. (See also *Figure 4—figure supplement 2*) (**D**) Images obtained during dialysis with MLCKp (0.1 µM). (**E**) Images from a cell in which CaM$_{1234}$ was co-expressed with Ca$_V$1.2-VN and Ca$_V$1.2-VC (see also *Figure 4—figure supplement 3*). (**F**) Relationship

domains, and is involved in CDI and CDF of Ca$_V$1.2 channels (*Eckert and Tillotson, 1981*). Dialysis with an intracellular solution containing the CaM inhibitory peptide MLCKp (0.1 µM) prevented Venus reconstitution during membrane depolarization (*Figure 4D,F,G*), without altering the rate of inactivation of I$_{Ca}$ (*Figure 4—figure supplement 3*). Expression of a mutant CaM that does not bind Ca$^{2+}$ (CaM$_{1234}$) also prevented Ca$_V$1.2-VN-Ca$_V$1.2-VC fusion upon membrane depolarization (*Figure 4E,F,G*). However, unlike MLCKp, CaM$_{1234}$ did slow the rate of I$_{Ca}$ inactivation (*Figure 4—figure supplement 3*), suggesting that two functionally distinct CaM molecules are involved in CDI and Ca$_V$1.2-to-Ca$_V$1.2 channel interactions.

Given the apparently essential role of Ca$^{2+}$/CaM, we next examined the importance of IQ and pre-IQ motif CaM-binding sites for Ca$_V$1.2-Ca$_V$1.2 channel interactions. To do so, we mutated critical residues in each segment and used bimolecular fluorescence complementation to evaluate changes in channel coupling ability. Ca$_V$1.2-VN and Ca$_V$1.2-VC channels with an I1654E mutation in their IQ domains were able to fuse upon membrane depolarization (*Figure 4—figure supplement 4*). This isoleucine residue is crucial for CaM binding to the IQ motif. Previous in vitro studies have reported that the I1654E (or the human homolog I1624E) mutation decreases the affinity of the IQ motif for CaM by ~100-fold (*Zühlke et al., 1999*, *2000*). Thus, our results suggest that CaM binding to the IQ-motif is not required for Ca$_V$1.2 channel coupling. To investigate the role of the pre-IQ motif in channel interactions, we exchanged a 33-amino-acid segment of the pre-IQ domain for 33 non-identical amino acids (*Figure 4—figure supplement 4A*). A similar segment exchange performed on human Ca$_V$1.2 channels expressed in tsA-201 cells was previously shown to impair channel clustering on the PM and reduce P$_o$ compared to wild-type (WT) channels (*Kepplinger et al., 2000*). Interestingly, the pre-IQ 'swap' mutation rendered Ca$_V$1.2-VN and Ca$_V$1.2-VC channels incapable of functional coupling (*Figure 4—figure supplement 4C*). Collectively, these results suggest that pre-IQ domains are required for Ca$^{2+}$/CaM-dependent Ca$_V$1.2 channel oligomerization. Additional credence is given to this finding by the previously published crystal structure of dimeric cardiac L-type Ca$^{2+}$ channels showing two pre-IQ helices bridged by two Ca$^{2+}$/CaMs (*Fallon et al., 2009*).

We next examined the spatial distribution of Ca$_V$1.2(pre-IQ swap) channels in tsA-201 cells

*Figure 4. Continued*

between membrane voltage and Venus reconstitution for each experimental condition. (**G**) Bar chart showing mean Venus fluorescence ($F/F_0$) ± SEM for each condition at −40 and +40 mV (*p < 0.05).

The following figure supplements are available for figure 4:

**Figure supplement 1**. Flash photolysis of caged $Ca^{2+}$ stimulates $Ca_V1.2$ interactions.

**Figure supplement 2**. $Ca_V1.2$ interactions are $Ca^{2+}$ dependent.

**Figure supplement 3**. $Ca^{2+}$ binding to distinct CaM pools regulates CDI.

**Figure supplement 4**. $Ca^{2+}$/CaM binding to the pre-IQ domain, and not the IQ domain, mediates channel coupling.

**Figure supplement 5**. $Ca_V1.2$ channel clustering is necessary but not sufficient for functional coupling.

co-transfected with mCherry-sec61β (to permit visualization of the ER). Surprisingly, $Ca_V1.2$(pre-IQ swap) channels still formed clusters in tsA-201 cells despite their inability to functionally interact (*Figure 4—figure supplement 5A*). Indeed, under identical imaging conditions (i.e., using the same fixative and TIRF penetration depth), $Ca_V1.2$(pre-IQ swap) channel cluster areas were not significantly different from those of WT channels. As noted above for WT channels, the $Ca_V1.2$(pre-IQ swap) channel cluster distribution was not limited to PM-ER junctions, even in the presence of JPH2 (*Figure 4—figure supplement 5A,B*). These results suggest that, while the physical proximity of $Ca_V1.2$ channels is necessary for channel interactions, it is not sufficient for functional coupling of the channels.

## $Ca_V1.2$ channel coupling augments activity

An important prediction of our findings is that the functional coupling produced by $Ca^{2+}$-induced $Ca_V1.2$-$Ca_V1.2$ channel interactions manifests as enhanced channel activity. To test this, we exploited the fact that Venus reconstitution is irreversible, recording $Ca_V1.2$ sparklets (*Wang et al., 2001*; *Navedo et al., 2005*) before and after $Ca^{2+}$-induced $Ca_V1.2$ coupling during membrane depolarization. Prior to membrane depolarization, $Ca_V1.2$ sparklet activity ($nP_s$) was low (0.002 ± 0.001). However, after depolarization, $nP_s$ increased ~40-fold (0.085 ± 0.022) and $Ca_V1.2$ sparklet density increased almost 10-fold (*Figure 5A,C,D* and *Video 2*; $n = 5$ cells). Membrane depolarization failed to significantly alter $nP_s$ or $Ca_V1.2$ sparklet density in cells co-expressing the $Ca^{2+}$-insensitive $CaM_{1234}$ mutant (*Figure 5B–D*; $n = 6$ cells). These data indicate that the post-depolarization augmentation of $Ca_V1.2$ sparklet activity resulted from a $Ca^{2+}$/CaM-dependent increase in the activity of previously active sites as well as the emergence of new, high-activity $Ca^{2+}$ sparklet sites that appear to lack CDI. These findings suggest that $Ca_V1.2$ channel interactions increase $Ca^{2+}$ influx and, consequently, total conductance by increasing the extent to which the channels are functionally coupled.

## $Ca_V1.2$ channel coupling is dynamic and transiently persistent

Having demonstrated that physical $Ca_V1.2$-to-$Ca_V1.2$ interactions functionally couple adjoining channels to enhance channel activity, we investigated the dynamics of these interactions. Bimolecular fluorescence complementation experiments are unable to provide information about channel-interaction dynamics, since Venus reconstitution is irreversible (*Kerppola, 2006*). Instead, we sought to detect fluorescence resonance energy transfer (FRET) between EGFP and red fluorescent protein (RFP)-tagged $Ca_V1.2$ channels as a function of $[Ca^{2+}]_i$. Flash photolysis of DMNP-EDTA (at −80 mV) induced a transient increase in $[Ca^{2+}]_i$ and enhanced the $Ca_V1.2$-RFP/$Ca_V1.2$-EGFP FRET ratio (FRETr; *Figure 6A*). The time courses of $[Ca^{2+}]_i$ and FRETr were similar (*Figure 6B*), with a time-to-peak of 605.8 ± 98.8 ms for $[Ca^{2+}]_i$ and 531.5 ± 57.5 ms for FRETr. The $[Ca^{2+}]_i$-FRETr relationship was sigmoidal, yielding a $FRETr_{1/2}$ at a $[Ca^{2+}]_i$ of approximately 250 nM (*Figure 6C*). Both $[Ca^{2+}]_i$ and FRETr traces followed triple exponential decay kinetics; for $[Ca^{2+}]_i$, the decay time constants (τ) for fast, intermediate and slow components were 0.62 ± 0.15, 2.19 ± 0.40 and 3.15 ± 0.26 s, respectively, and the corresponding values for FRETr were 0.64 ± 0.48, 1.09 ± 0.97 and 3.71 ± 0.73 s ($n = 5$). The time to decay to 50% of the peak ($T_{50\%}$) was 2.65 ± 0.22 s for $[Ca^{2+}]_i$ and 0.63 ± 0.23 s for FRETr. The complex multi-component decay kinetics of each trace reflects the multiple $Ca^{2+}$ binding sites and affinities of the two lobes of CaM for $Ca^{2+}$ (*Faas et al., 2011*; *Faas and Mody, 2012*).

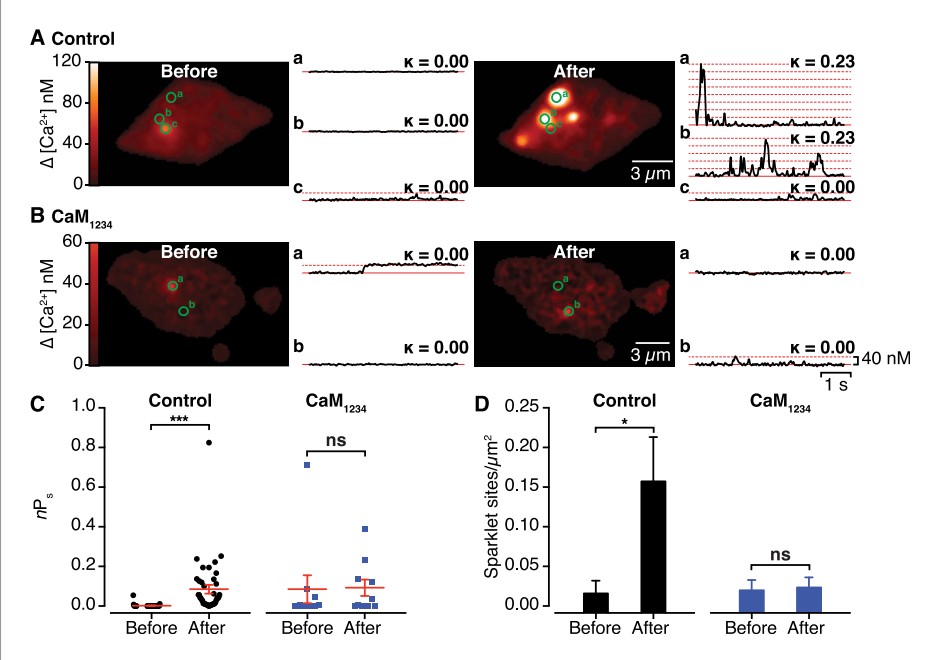

**Figure 5**. Effects of interactions between $Ca_V1.2$ channel C-termini on channel activity. (**A** and **B**) Calibrated TIRF images (see also **Video 2**) of representative control tsA-201 cells (**A**) and tsA-201 cells expressing the $Ca^{2+}$-insensitive $CaM_{1234}$ mutant (**B**). In both cases, cells expressed $Ca_V1.2$-VN and $Ca_V1.2$-VC and were loaded with Rhod-2 via the patch pipette. Cells were held at −80 mV during sparklet recordings before (*left*) and after (*right*) depolarization to +60 mV. The time course of $[Ca^{2+}]_i$ for each sparklet site (denoted by green circles) before and after depolarization is shown to the right of each image. (**C** and **D**) Scatter plots of sparklet activity (**C**; $nP_s$; ***p < 0.001) and sparklet site density (**D**; *p = 0.03), before and after depolarization in control (*n* = 5) and $CaM_{1234}$-expressing (*n* = 6) cells. ns, not significant.

Concurrent recordings of currents and FRETr during membrane depolarization in the presence of $Ca^{2+}$ (with 10 mM EGTA in the pipette solution) showed that depolarization to +10 mV evoked a transient increase in FRETr (**Figure 6D**). The increase in FRETr appeared biphasic, with a large amplitude spike followed by a lower level plateau that was sustained for the duration of the increase in $[Ca^{2+}]_i$. These data suggest that $Ca_V1.2$-$Ca_V1.2$ interactions are dynamic and regulated by local and global changes in $[Ca^{2+}]_i$. Notably, the increase in FRETr persisted after the $Ca^{2+}$ current had decayed to baseline, indicating that channels remained coupled for a time in the absence of a stimulus.

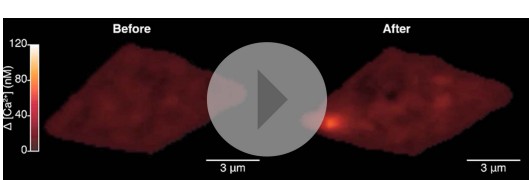

**Video 2.** $Ca^{2+}$ sparklet activity and site density are augmented by $Ca_V1.2$ channel interactions. Stacks of 2D images acquired at a frame rate of 100 Hz from a representative, Rhod-2–dialyzed tsA cell expressing $Ca_V1.2$-VN and $Ca_V1.2$-VC, held at −80 mV, before (*left*) and after (*right*) depolarization.

## $Ca_V1.2$ channel coupling facilitates $I_{Ca}$

An explicit implication of our results is that physical $Ca_V1.2$-$Ca_V1.2$ interactions are critical for CDF in ventricular myocytes. To test this hypothesis, we investigated whether $Ba^{2+}$ permeation and inhibition of CaM with MLCKp—both of which prevent $Ca_V1.2$-$Ca_V1.2$ interaction (see above)—decreases or eliminates CDF in ventricular myocytes. Because $Ca^{2+}$/CaM-dependent kinase II (CaMKII) has been implicated in $Ca^{2+}$ current facilitation (**Anderson et al., 1994**; **Xiao et al., 1994**; **Yuan and Bers, 1994**), we included 10 mM EGTA in the patch pipette to maintain global

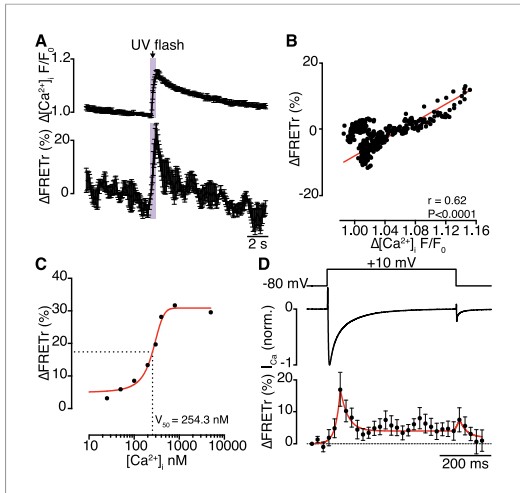

**Figure 6**. Interactions of $Ca_V1.2$ C-termini occur dynamically with $Ca^{2+}$ influx and are transiently persistent. (**A**) Time course of the percent change in FRETr (*bottom*) evoked by flash photolysis of caged $Ca^{2+}$ (purple box). Experiments were performed at a holding potential of −80 mV with zero EGTA or BAPTA; thus, the change in $[Ca^{2+}]_i$ ($F/F_0$; *top*) was global. Averaged traces ($n = 5$ cells) and error bars showing SEM at each sampling point. (**B**) Correlation between changes in FRETr and global $[Ca^{2+}]_i$ produced by caged $Ca^{2+}$ photolysis, showing that increases in $[Ca^{2+}]_i$ were accompanied by an increase in FRETr ($p < 0.0001$). (**C**) Plot of the percent change in FRETr vs $[Ca^{2+}]_i$ (nM). Data ($n = 8$ cells) were fit to a Boltzmann Sigmoidal function (solid red line) with $V_{50} = 254.3$ nM (black dashed line). (**D**) Step depolarization (*top*) in the presence of 10 mM EGTA produced local $[Ca^{2+}]_i$ elevation and inactivating $I_{Ca}$ (*middle*). Increased channel interactions, represented as the percent change in FRETr (*bottom*; averaged from $n = 6$ cells), were detected at the onset of depolarization and repolarization during the peak and tail currents. Dashed line shows FRETr baseline level.

$[Ca^{2+}]_i < 50$ nM, which is below the threshold ($>200$ nM) for activation of this kinase (*Miller and Kennedy, 1986*). Experiments were performed in cells dialyzed with 100 nM or 1 μM MLCKp. The rationale for using these two concentrations of MLCKp is that, whereas 100 nM MLCKp inhibits CaM (*Török and Trentham, 1994*; *Török et al., 1998*), we found that it does not change CaMKII activity ($p < 0.05$). However, we found that 1 μM MLCKp inhibits CaM and eliminates CaMKII activity. Thus, using these two MLCKp concentrations, we can selectively dissect the contribution of CaM and locally activated (i.e., near the channel pore) CaMKII.

We used a three-step protocol (*Figure 7A*) to record facilitated $Ca_V1.2$ currents as previously described (*Poomvanicha et al., 2011*). Briefly, cells were held at −80 mV and $I_{Ca}$ was elicited with a 200-ms control pulse ($V_1$) to 0 mV. Cells were held again at −80 mV for 10 s followed by a 200-ms prepulse to +80 mV ($V_{pre}$). In light of our FRET results suggesting that $Ca_V1.2$ channels remain coupled for a time in the absence of a stimulus, we held the cells at −80 mV for a variable interpulse interval of 0.1–1.6 s before beginning the second 200-ms test pulse ($V_2$) to 0 mV. Fast $Na^+$ currents were inactivated with a 50-ms step to −40 mV applied prior to each control ($V_1$) or test pulse ($V_2$). The ratio of $I_{Ca}$ resulting from test and control pulses ($I_2/I_1$) was used as a measure of facilitation ($I_2/I_1 > 1$) and recovery from inactivation ($I_2/I_1 \approx 1$).

In control cells (no peptide, 2 mM external $Ca^{2+}$), this protocol induced $I_{Ca}$ facilitation (i.e., $I_2/I_1 > 1$) (*Figure 7B*). Note, however, that the magnitude of $I_{Ca}$ facilitation varied with interpulse duration, reaching a peak at a $V_{pre}$-$V_2$ interval of 0.3 s. Longer $V_{pre}$-$V_2$ intervals induced progressively less $I_{Ca}$ facilitation. Indeed, $I_2/I_1$ was statistically indistinguishable from 1 at interpulse durations longer than 1.2 s, suggesting no $I_{Ca}$ facilitation at stimulation rates >0.8 Hz.

In the presence of extracellular $Ca^{2+}$, intracellular dialysis with 0.1 μM MLCKp suppressed $I_{Ca}$ facilitation at interpulse intervals of 0.1–0.3 s ($n = 10$ for 0.1 s intervals; $n = 5$ for 0.2 and 0.3 s intervals), but control levels of facilitation returned with intervals of 0.4–1 s ($n = 5$–9, $p < 0.05$; *Figure 7B*). Dialysis with a 10-fold higher concentration of MLCKp (1 μM) eliminated $I_{Ca}$ facilitation at all interpulse intervals ($n = 4$; *Figure 7B*). Finally, superfusion of ventricular myocytes with $Ba^{2+}$ was as effective as dialysis with 1 μM MLCKp in preventing $I_{Ca}$ facilitation (*Figure 7B–D*).

We performed a detailed analysis of our data to gain more insights into the relationship between $Ca_V1.2$ channel coupling and $I_{Ca}$ facilitation during high frequency stimulation. The amplitude of $I_2$ in the paired pulse facilitation would depend, in part, on the degree of recovery from inactivation. A previous study has determined that the time course of $I_{Ca}$ recovery from inactivation follows a single exponential function with a $\tau_{recovery}$ of about 375 ms (*Blaich et al., 2010*). Our FRET data in *Figure 6D* suggest that the number of coupled $Ca_V1.2$ channels will fade exponentially with an estimated $\tau_{decoupling}$ of 333 ms upon repolarization. In *Figure 7E* (*left*), we show a simulation of the time-course of $I_{Ca}$ recovery and $Ca_V1.2$ channel decoupling (*middle*) during repolarization using exponential

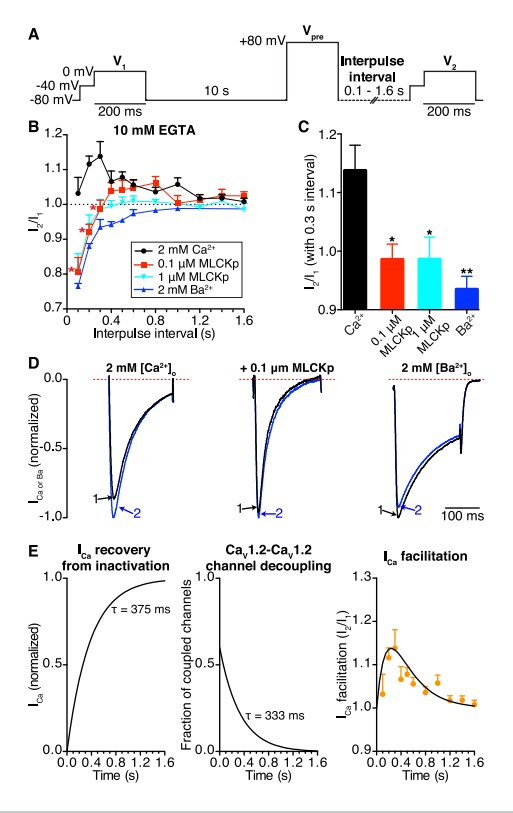

**Figure 7**. $Ca_V1.2$-to-$Ca_V1.2$ channel coupling is critical for $I_{Ca}$ facilitation in cardiomyocytes. (**A**) Voltage protocol used to evoke $I_1$ and $I_2$ in ventricular myocytes. (**B**) Line chart summarizing the current amplitude ratio ($I_2/I_1$) at 0 mV for each condition over the range of interpulse intervals from 0.1 to 1.6 s. (**C**) Bar chart summarizing the current-amplitude ratio ($I_2/I_1$) with a fixed interpulse interval of 300 ms for each condition. (**D**) Normalized whole-cell currents evoked by the protocol in (**A**), with 2 mM $[Ca^{2+}]_o$ as the charge carrier without (*left*) or with (*middle*) intracellular dialysis of 0.1 µM MLCKp (*middle*). The currents on the right were recorded with 2 mM $[Ba^{2+}]_o$ as the charge carrier. Data are shown as means + SEM (*p < 0.05, **p < 0.01 vs control (2 mM $[Ca^{2+}]_o$)). (**E**) Simulated time-course of $I_{Ca}$ recovery from inactivation (*left*), $Ca_V1.2$-$Ca_V1.2$ channel cluster disassembly (*middle*), and $I_{Ca}$ facilitation (solid black line; *right*). Recovery and channel decoupling curves are single exponential functions with time constants ($\tau$) of 375 ms and 333 ms, respectively. The $I_{Ca}$ facilitation curve is the product of the recovery and decoupling functions. For comparison, the experimental facilitation data (orange circles) collected with 2 mM $[Ca^{2+}]_o$ as the charge carrier is plotted alongside the simulated $I_{Ca}$ facilitation data. The amplitude of the decoupling function was scaled to fit the facilitation data.

functions with these $\tau_{decoupling}$ and $\tau_{recovery}$ values. Note that the time-course of $I_{Ca}$ facilitation data obtained from myocytes superfused with $Ca^{2+}$ is well described by the product of the recovery and decoupling functions (*Figure 7E*; *right*). This analysis is consistent with a model in which $I_{Ca}$ facilitation during high frequency stimulation is directly proportional to the number of coupled channels and the number of channels available for activation.

These data support the view that $I_{Ca}$ facilitation in ventricular myocytes depends on $Ca^{2+}$ influx and CaM, but has CaMKII-dependent and independent components. In combination with our FRET and split-Venus data, these findings suggest that $Ca^{2+}$ augments $Ca_V1.2$ channel activity at least in part by increasing the number of functionally coupled channels.

## Discussion

Our results support a new model for $Ca_V1.2$ channel function. An illustration of our proposed model for coupled gating of $Ca_V1.2$ channels is shown in *Figure 8A–D*. In our formulation, CDI and CDF are interrelated processes, both dependent on the coupling state of adjacent $Ca_V1.2$ channels. The C-terminal tail of the channel serves a dual role: inducing the functional coupling of adjacent channels via protein-to-protein interactions and regulating channel open probability. The physical interaction between clustered $Ca_V1.2$ channels is tightly regulated by local and global $[Ca^{2+}]_i$. The cascade of events that culminates in the coupling of $Ca_V1.2$ channels during an action potential begins with the gating of an individual channel within a cluster. The resulting $Ca_V1.2$ sparklet induces the binding of $Ca^{2+}$ to CaM in the pre-IQ domain of the channel, which promotes physical interactions between contiguous channels. This increases the activity of adjoined channels, elevating local $[Ca^{2+}]_i$. As individual channels within a cluster undergo VDI and CDI and close, $[Ca^{2+}]_i$ decreases and coupled channels disassemble. This, in turn, decreases channel opening probability and terminates $Ca^{2+}$ flux. Thus, the overall activity of $Ca_V1.2$ channels within a cluster depends on the number of channels that form dimers or higher-order oligomers.

Our results have profound implications for current models of cardiac EC coupling as they provide an answer to a long-standing question in

the field: What are the mechanisms that allow the simultaneous, coordinated opening of multiple $Ca_V1.2$ channels near the jSR? EC coupling starts with membrane depolarization. According to our model, membrane depolarization increases the probability of $Ca_V1.2$ sparklet occurrence. $Ca_V1.2$

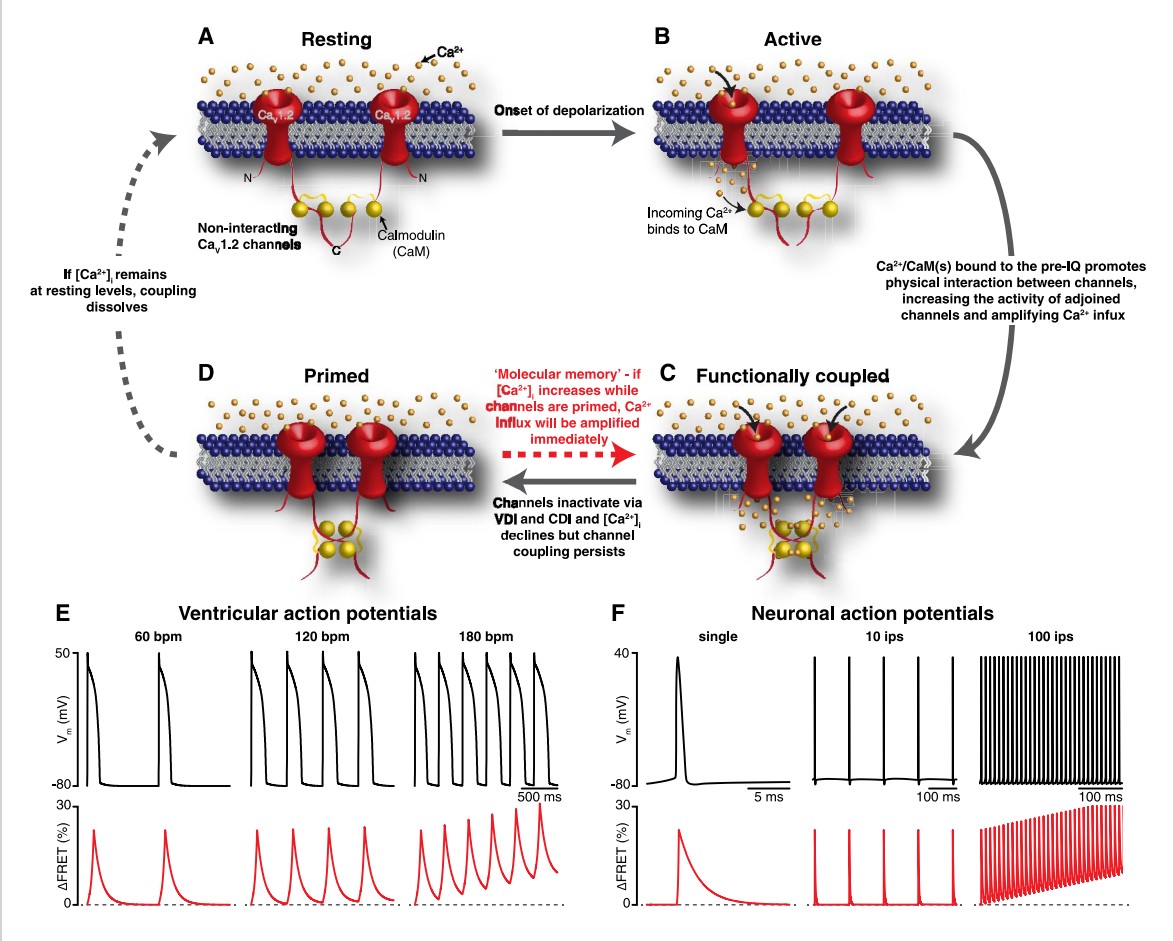

**Figure 8**. Mechanism and proposed model for the functional coupling of Ca$_V$1.2 channels. (**A**) Ca$_V$1.2 channels are arranged into clusters in the PM of excitable cells; for simplicity, a cluster of two channels is shown. At the resting membrane potential (e.g., −80 mV), [Ca$^{2+}$]$_i$ and Ca$_V$1.2 $P_o$ are low; hence, the majority of Ca$_V$1.2 channels are non-interacting. (**B**) During an action potential, the PM becomes depolarized, increasing the $P_o$ of independently gating Ca$_V$1.2 channels. Ca$^{2+}$ flows into the cell through these active channels, producing an elevation in local [Ca$^{2+}$]$_i$ and increasing Ca$^{2+}$ binding to CaM. (**C**) Ca$^{2+}$/CaM binding to the C-terminal pre-IQ domain of the Ca$_V$1.2 channel promotes physical interactions between adjacent channels. This functional coupling increases the activity of adjoined channels and thus amplifies Ca$^{2+}$ influx. (**D**) Ca$_V$1.2 channels undergo VDI and CDI, and [Ca$^{2+}$]$_i$ declines once more. However, the channels remain coupled in a 'primed', non-conducting state for a finite time. If the membrane is depolarized again when the channels are still primed, the amplification of Ca$^{2+}$ influx will be immediate; otherwise, if [Ca$^{2+}$]$_i$ remains at resting levels beyond the lifetime of the primed state, the coupling dissolves and the cycle begins again. (**E**) and (**F**) show proposed rate-dependent changes in Ca$_V$1.2 channel coupling in ventricular myocytes and neurons, respectively. *Top*: Simulated ventricular and neuronal action potentials are depicted at low, intermediate, and high firing rates. *Bottom*: The accompanying dynamic change in Ca$_V$1.2 channel coupling (reflected by FRET changes between adjacent channels). bpm, beats per minute; ips, impulses per second.

sparklets elevate local [Ca$^{2+}$]$_i$, thereby increasing the number of channels that form dimers or higher-order oligomers. This physical coupling increases the probability of synchronous, coincident openings of channels within a cluster (∼5–10 channels) to reliably activate nearby RyRs through Ca$^{2+}$-induced Ca$^{2+}$ release (*Inoue and Bridge, 2003*; *Sobie and Ramay, 2009*). Accordingly, the strength of cardiac contraction would depend, at least in part, on the number of physically and functionally coupled Ca$_V$1.2 channels.

Our data further suggest that the degree of Ca$_V$1.2-Ca$_V$1.2 coupling varies within the physiological range of [Ca$^{2+}$]$_i$ reached in ventricular myocytes during a cardiac cycle. We found that Ca$_V$1.2 channel coupling is dynamic and has an apparent $K_d$ of ∼250 nM. While the model assumes that Ca$_V$1.2-to-Ca$_V$1.2 coupling is initiated by a Ca$^{2+}$ sparklet, it could also be induced by a Ca$^{2+}$ spark resulting from the opening of a small cluster of closely apposed RyRs. This could induce local Ca$_V$1.2-to-Ca$_V$1.2 coupling, even if transiently, priming these channels for opening during the

action potential. Note, however, that because $Ca^{2+}$ spark activity is very low during diastole, the number of coupled $Ca_V1.2$ channels would likely be low. However, at the peak level of the $[Ca^{2+}]_i$ transient (~700 nM) during EC coupling (*Santana et al., 2002*), the cell would have reached a maximal level of $Ca_V1.2$-to-$Ca_V1.2$ coupling.

An important finding in our study is that, while $Ca_V1.2$ channel coupling fades as $[Ca^{2+}]_i$ decreases, it persists longer than the current that evoked it. This is important because $Ca_V1.2$ channel activity remains elevated for as long as the channels remain coupled. Thus, by outlasting the $[Ca^{2+}]_i$ signal that evoked it, $Ca_V1.2$ channel coupling acts as a type of 'molecular memory' that might serve to augment $Ca^{2+}$ influx during repetitive membrane depolarization. As a consequence, an increase in action potential frequency can enhance $Ca^{2+}$ influx via two mechanisms: first, it can increase $[Ca^{2+}]_i$, thereby increasing $Ca_V1.2$ channel coupling; and second, if an AP arrives while a subpopulation of $Ca_V1.2$ channels remain coupled, which could occur even if $[Ca^{2+}]_i$ had decreased to basal levels owing to the molecular memory phenomenon, it would encounter a cell with a higher number of coupled—that is, more active—channels (see *Figure 8E,F*). This molecular memory might also manifest itself as $Ca_V1.2$ current facilitation (*Marban and Tsien, 1982*; *Argibay et al., 1988*; *Gurney et al., 1989*). In the case of cardiac muscle, $Ca_V1.2$ current facilitation augments contractile force during increases in heart rate (*Pieske et al., 1999*). In neurons, facilitation of $Ca_V1.2$ channels has been suggested to contribute to an intrinsic amplification of synaptic current and the enhancement of neuronal excitability (*Powers and Binder, 2001*; *Striessnig et al., 2014*).

Our data support the view that $Ca^{2+}$ current facilitation in ventricular myocytes requires $Ca^{2+}$ influx and CaM, but has CaMKII-dependent and -independent components. The latter likely involve $Ca^{2+}$/CaM-induced $Ca_V1.2$-$Ca_V1.2$ channel coupling. Future experiments should investigate if CaMKII facilitates $Ca^{2+}$ current in neurons and cardiac muscle by a similar mechanism.

Dynamic, $[Ca^{2+}]_i$-dependent $Ca_V1.2$ channel coupling could have pathological repercussions. For example, coupling of a mutant channel with a higher intrinsic open probability to a WT channel could increase $Ca^{2+}$ influx. One condition associated with aberrantly high $Ca_V1.2$ channel gating is Timothy syndrome (TS), also known as long-QT syndrome 8. TS is an autosomal-dominant, multi-organ disorder caused by de novo gain-of-function missense mutations in exon 8 or an alternatively spliced exon 8A of $Ca_V1.2$ (*Splawski et al., 2004*, *2005*). Forced physical interactions between $Ca_V1.2$ channels have an intriguing effect on adjoined channels: fusion of intrinsically hyperactive $Ca_V1.2$-TS channels to WT channels induces these latter channels to function like TS channels. These findings have important implications. If TS channels can form stable interactions with neighboring WT channels in TS patients, then these mutant channels, which constitute only ~23% of the total cardiac $Ca_V1.2$ population, could have a disproportionally large effect on $Ca^{2+}$ influx. We previously tested this idea in ventricular myocytes expressing our fusible $Ca_V1.2$ channels and found that fusion of TS channels with WT channels led to the development of arrhythmogenic spontaneous SR $Ca^{2+}$ release events in addition to increasing the amplitude of $[Ca^{2+}]_i$ transients and contractions (*Dixon et al., 2012*). These findings support the hypothesis that physical interactions between the C-termini of TS and WT channels produce a disproportionally large $Ca^{2+}$ influx that ultimately induces arrhythmogenic changes in $[Ca^{2+}]_i$. Consistent with this, we found that the relationship between the level of $Ca_V1.2$-TS channel expression and the probability of a $Ca^{2+}$ wave is non-linear, suggesting that even low levels of these channels are sufficient to induce maximal changes in $[Ca^{2+}]_i$ (*Drum et al., 2014*).

Mutations in CaM have recently been linked to severe forms of long-QT syndrome, which are associated with life-threatening arrhythmias that occur very early in life (*Limpitikul et al., 2014*). Expression of these CaMs increases $Ca_V1.2$ activity and open times, effects similar to those produced by the TS mutation. Because CaM regulates many other $Ca^{2+}$ channel subtypes, including those that predominate in neurons, these mutant CaMs could lead to a multi-system disorder similar to TS. An important direction for future experiments will be to investigate whether $Ca_V1.2$ channel dysfunction is associated with aberrant TSTS or TS-WT $Ca_V1.2$ channel coupling or coupling of channels with mutant CaM to channels with WT CaM.

In summary, our study demonstrates that dynamic, $Ca^{2+}$-driven physical interactions among clustered $Ca_V1.2$ channels lead to cooperative gating of adjacent channels and enhanced $Ca^{2+}$ influx. The physical proximity afforded by clustering of $Ca_V1.2$ channels is necessary, but not sufficient, for functional coupling of the channels and occurs whether the channels are functionally coupled or not. Future studies should investigate the mechanisms that dictate the clustered arrangement of $Ca_V1.2$

channels. It is likely that cooperative $Ca_V1.2$ channel gating also plays an important role in physiological functions as diverse as neuronal excitability and rate-dependent increases in cardiac contraction, as well as pathological conditions such as long-QT syndrome.

## Materials and methods

### Isolation of ventricular myocytes

WT C57BL/6 mice were euthanized with a single lethal dose of sodium pentobarbital delivered via intraperitoneal injection, as approved by the University of Washington Institutional Animal Care and Use Committee (IACUC). Ventricular myocytes were isolated as previously described (*Dixon et al., 2012*). Briefly, the heart was excised and rinsed with cold 150 µM EGTA digestion buffer containing 130 mM NaCl, 5 mM KCl, 3 mM Na-pyruvate, 25 mM HEPES, 0.5 mM $MgCl_2$, 0.33 mM $NaH_2PO_4$, and 22 mM glucose. The aorta was cannulated for Langendorff perfusion, and the coronary arteries were subsequently perfused with warmed (37°C) 150 µM EGTA digestion buffer until they were cleared of blood. The perfusate was then switched to digestion buffer (no EGTA) supplemented with 50 µM $CaCl_2$, 0.04 mg/ml protease (XIV), and 1.4 mg/ml collagenase (type 2; Worthington Biochemical, Lakewood, NJ) for 8–10 min. The ventricles were then cut away from the atria, sliced, and placed in 37°C digestion buffer supplemented with 0.96 mg/ml collagenase, 0.04 mg/ml protease, 100 µM $CaCl_2$, and 10 mg/ml bovine serum albumen (BSA). Gentle agitation was applied using a transfer pipette until the ventricles dissociated. The cells were then allowed to pellet by gravity for 15–20 min, after which they were washed in enzyme-free digestion buffer supplemented with 10 mg/ml BSA and 250 µM $CaCl_2$, pelleted once more, and finally resuspended at room temperature in Tyrode's solution containing 140 mM NaCl, 5 mM KCl, 10 mM HEPES, 10 mM glucose, 2 mM $CaCl_2$, and 1 mM $MgCl_2$; pH was adjusted to 7.4 with NaOH.

### Plasmid constructs and tsA-201 cell transfection

tsA-201 cells (Sigma–Aldrich, St. Louis, MO) were cultured in Dulbecco's Modified Eagle Medium (DMEM; Gibco-Life Technologies, Grand Island, NY) supplemented with 10% fetal bovine serum (FBS) and 1% penicillin/streptomycin at 37°C in a humidified 5% $CO_2$ atmosphere and passaged every 3–4 day. Cells were transiently transfected at ~70% confluence using jetPEI (Polyplus Transfection, New York, NY) transfection reagent and plated onto the appropriate coverglass ~12 hr before experiments. For super-resolution imaging and photobleaching experiments, cells were plated onto 22 mm no. 1.5 coverslips (Thermo Fisher Scientific, Waltham, MA). For all other experiments, cells were plated onto 25 mm no. 1 coverslips (Thermo Fisher Scientific). Plasmids used in this study include pcDNA clones of the pore-forming subunit of rabbit $Ca_V1.2$ ($\alpha_{1c}$) and rat auxillary subunits $Ca_V\alpha_2\delta$ and $Ca_V\beta_3$ (kindly provided by Dr Diane Lipscombe; Brown University, Providence, RI). Standard PCR techniques were used to fuse the carboxyl tail of $Ca_V1.2$ to different proteins depending on the experimental approach: for FRET experiments, to EGFP or tagRFP; for bimolecular fluorescence complementation, to either the N-fragment (VN) or the C-fragment (VC) of the Venus protein (27097, 22011; Addgene, Cambridge, MA) (*Kodama and Hu, 2010*); for photobleaching experiments, to the monomeric GFP variant GFP(A206K) (*Zacharias et al., 2002*), kindly provided by Dr Eric Goaux (Vollum Institute, Portland, OR). $Ca^{2+}$-insensitive $CaM_{1234}$ was a gift from Dr Johannes Hell (UC Davis, CA). Mutant rabbit $Ca_V1.2(I1654E)$, analogous to human I1624E (*Zühlke et al., 1999*), was used in experiments designed to elucidate the role of the IQ motif in channel interactions. This single point mutation (I1672E) was introduced using the QuikChange II XL Site-Directed Mutagenesis kit (Agilent Technologies, Santa Clara, CA). The role of the pre-IQ motif in channel interactions was examined by exchanging a 33-amino-acid segment, as illustrated in *Figure 4—figure supplement 4A*. The resultant $Ca_V1.2(pre-IQ swap)$ was used in whole-cell patch-clamp and bimolecular fluorescence complementation experiments. The general ER marker mCherry-Sec61β was a gift from Gia Voeltz (Addgene plasmid # 49155). Finally, JPH2-GFP was used to tether the ER to the PM in tsA-201 cells.

### Electrophysiology

$Ca^{2+}$ currents were recorded in the whole-cell voltage-clamp or cell-attached patch configurations using borosilicate patch pipettes with resistances of 3–6 µΩ for tsA cells and 2–3 µΩ for cardiomyocytes. Currents were sampled at a frequency of 20 kHz, low-pass–filtered at 2 kHz using an Axopatch 200B amplifier, and acquired using pClamp 10.2 software (Molecular Devices, Sunnyvale, CA).

All membrane potentials referred to herein have been corrected for liquid junction potential. All experiments were performed at room temperature (22–25°C).

For whole-cell current recordings from tsA-201 cells, pipettes were filled with a Cs-based internal solution containing 87 mM Cs-aspartate, 20 mM CsCl, 1 mM MgCl$_2$, 10 mM HEPES, 10 mM EGTA and 5 mM MgATP, adjusted to pH 7.2 with CsOH. In experiments requiring dialysis of the calmodulin inhibitory peptide MLCKp (EMD Millipore, Darmstadt, Germany; 208735), this chemical was added to the pipette solution. Cells were continuously superfused throughout experiments with our regular external solution containing 5 mM CsCl, 10 mM HEPES, 10 mM glucose, 113 mM NMDG, 1 mM MgCl$_2$ and 20 mM CaCl$_2$, adjusted to pH 7.4 with HCl. For experiments in which Ba$^{2+}$ was used as the charge carrier in place of Ca$^{2+}$, the perfusate contained 5 mM CsCl, 10 mM HEPES, 10 mM glucose, 140 mM NMDG, 1 mM MgCl$_2$ and 2 mM BaCl$_2$, adjusted to pH 7.4 with HCl. Current-voltage relationships were obtained by subjecting cells to a series of 300-ms depolarizing pulses from a holding potential of −80 mV to test potentials ranging from −60 to +80 mV. The voltage dependence of conductance was obtained by converting the resultant currents to conductances using the equation, G = $I_{Ca}$/[test pulse potential − reversal potential of $I_{Ca}$], normalizing (G/Gmax), and plotting conductance vs the test potential.

Cell-attached patch, single-channel currents ($i_{Ca}$) were recorded from tsA-201 cells superfused with high K$^+$ solution to fix the membrane potential at ∼0 mV. The solution had the following composition: 145 mM KCl, 2 mM MgCl$_2$, 0.1 mM CaCl$_2$, 10 mM HEPES and 10 mM glucose; pH was adjusted to 7.3 with KOH. Pipettes were filled with a solution containing 10 mM HEPES and either 110 mM CaCl$_2$ or 110 mM BaCl$_2$; pH was adjusted to 7.2 with CsOH. The dihydropyridine agonist BayK-8644 (500 nM) was included in the pipette solution to promote longer channel open times. A voltage-step protocol from a holding potential of −80 mV to a depolarized potential of −30 mV was used to elicit currents. The single-channel event-detection algorithm of pClamp 10.2 was used to measure single-channel opening amplitudes and $n$P$_o$, and to construct all-points histograms.

To record I$_{Ca}$ from isolated ventricular myocytes, cells were initially perfused with Tyrode's solution. Once the whole-cell configuration was successfully established, the external solution was replaced with one containing 5 mM CsCl, 10 mM HEPES, 10 mM glucose, 140 mM NMDG, 1 mM MgCl$_2$ and either 2 mM CaCl$_2$ or 2 mM BaCl$_2$, adjusted to pH 7.4 with HCl. The pipette was filled with the Cs-based internal solution described above. Facilitation of I$_{Ca}$ was measured using a triple-pulse protocol consisting of two identical test pulses (V$_1$ and V$_2$), separated by a conditioning pulse to +80 mV (V$_{pre}$), as previously described (*Poomvanicha et al., 2011*) and illustrated in *Figure 7A*. The currents elicited by V$_1$ and V$_2$ were referred to as I$_1$ and I$_2$, and the ratio between their peaks (I$_2$/I$_1$) was used as a measure of facilitation.

## Recording of Ca$^{2+}$ sparklets

To measure Ca$^{2+}$ sparklets in tsA-201 cells or ventricular myocytes, cells were patch-clamped in whole-cell mode and held at a hyperpolarized potential of −80 mV to increase the driving force for Ca$^{2+}$ entry. Sub-sarcolemmal Ca$^{2+}$ signals were monitored by dialyzing cells with 200 µM Rhod-2 via the patch pipette and continuously perfusing them with the external solution described above containing 20 mM CaCl$_2$ and 10 mM EGTA, a relatively slow Ca$^{2+}$ buffer. With this buffer/indicator combination, Ca$^{2+}$ entering the cell via membrane Ca$_v$1.2 channels binds to the relatively fast Ca$^{2+}$ indicator Rhod-2 to generate a fluorescent signal, and the excess EGTA rapidly buffers Ca$^{2+}$, restricting the signal to the point of entry. Sub-sarcolemmal Ca$^{2+}$ signals (Ca$^{2+}$ sparklets) were captured using a through-the-lens TIRF microscope built around an Olympus IX-70 inverted microscope equipped with an oil-immersion ApoN 60×/1.49 NA TIRF objective and an Andor iXON CCD camera. Images were acquired at 100 Hz using TILLvisION imaging software (TILL Photonics, FEI, Hillsboro, OR). Sparklets were detected and analyzed using custom software written in MATLAB (*Source code 1*). Rhod-2 fluorescence signals were converted to Ca$^{2+}$ concentration units using the F$_{max}$ equation (*Maravall et al., 2000*). The activity of Ca$^{2+}$ sparklets was determined by calculating the $n$P$_s$ of each Ca$^{2+}$ sparklet site, where $n$ is the number of quantal levels and P$_s$ is the probability that a quantal Ca$^{2+}$ sparklet event is active. A detailed description of this analysis can be found in Navedo et al. (*Navedo et al., 2005*, *2006*).

## Coupled Markov chain model

The degree of coupling between single $Ca_V1.2$ channels or $Ca^{2+}$ sparklet sites was assessed by further analyzing single-channel and sparklet recordings using a binary coupled Markov chain model (*Source code 2*), as first described by *Chung and Kennedy (1996)* and previously employed by our group (*Navedo et al., 2010*; *Cheng et al., 2011*; *Dixon et al., 2012*). The custom program, written in the MATLAB language, assigns a coupling-coefficient (κ) to each record, where κ can range from 0 (purely independently gating channels) to 1 (fully coupled channels). Elementary event amplitudes were set at 0.5 pA for $i_{Ca}$, 1.5 pA for $i_{Ba}$, and 38 nM for $Ca^{2+}$ sparklets.

## Immunocytochemistry and super-resolution microscopy

Transfected tsA-201 cells expressing the relevant $Ca_V1.2$ channel constructs were plated onto poly-L-lysine–coated #1.5 coverslips (Thermo Fisher Scientific) the day before fixation. For immunostaining, cells were fixed by incubating for 10 min in ice-cold methanol, then washed with PBS and blocked for 1 hr at room temperature in 50% SEA BLOCK (Thermo Fisher Scientific) and 0.5% vol/vol Triton X-100 in PBS (blocking buffer). The pore-forming subunit of $Ca_V1.2$ was probed with a rabbit polyclonal primary antibody (anti-CNC1; kindly provided by Drs William Catterall and Ruth Westenbroek [*Hulme et al., 2003*]), diluted to 5 µg/ml in diluted blocking buffer (20% SEA BLOCK, 0.5% Triton X-100), by incubating overnight at 4°C. The following morning, cells were washed extensively, receiving three washes with PBS and five washes with diluted blocking buffer. Cells were then incubated for 1 hr at room temperature with Alexa Fluor 647-conjugated donkey anti-rabbit secondary antibody (2 µg/ml; Molecular Probes–Life Technologies) in diluted blocking buffer. Cells were finally washed thoroughly with PBS and mounted for imaging. Native $Ca_V1.2$ channels in freshly isolated adult ventricular myocytes were immunostained in an identical manner except that plating procedures differed. Specifically, myocytes were plated onto laminin and poly-L-lysine–coated #1.5 coverslips and allowed to adhere for 1 hr before fixation.

For double-labeling experiments used to examine the co-localization of $Ca_V1.2$ channels and the ER, tsA-201 cells transfected with $Ca_V1.2$ channels (WT or pre-IQ swap mutants) and mCherry-Sec61β were fixed in 3% paraformaldehyde and 0.1% glutaraldehyde in PBS for 10 min followed by extensive washing in PBS and reduction for 5 min in ∼0.1% sodium borohydride in water to reduce background fluorescence. Cells were washed and blocked as described above and were then incubated in primary antibody solution containing 5 µg/ml rabbit anti-CNC1 and 2 µg/ml rat monoclonal anti-mCherry (Life Technologies) for 1 hr at room temperature. Excess primary antibody was removed by three washes with PBS and five washes with diluted blocking buffer. Cells were then incubated for 1 hr at room temperature with Alexa Fluor 647-conjugated donkey anti-rabbit and Alexa Fluor 568-conjugated chicken anti-rat secondary antibodies (2 µg/ml each; Molecular Probes–Life Technologies) in diluted blocking buffer. Cells were finally washed thoroughly with PBS and mounted for imaging. In some experiments, tsA-201 cells co-expressing JPH2-GFP were fixed and stained as per the double-staining protocol described above. JPH2-GFP was not immunostained, but simply imaged in TIRF mode by exciting the GFP tag.

Coverslips were mounted with MEA-GLOX (for double staining) or β-ME-GLOX imaging buffer on glass depression slides (neoLab, Heidelberg, Germany) and sealed with Twinsil (Picodent, Wipperfürth, Germany). The imaging buffers contained TN buffer (50 mM Tris pH 8.0, 10 mM NaCl), GLOX oxygen scavenging system (0.56 mg/ml glucose oxidase, 34 µg/ml catalase, 10% wt/vol glucose), and either 100 mM MEA (cysteamine) or 142 mM 2-mercaptoethanol (β-ME). Excess imaging buffer was blotted away before application of the Twinsil sealant. This is particularly important with the β-ME-GLOX imaging buffer as the Twinsil will not set if it contacts this buffer.

GSD super-resolution images of $Ca_V1.2$ channels in fixed ventricular myocytes or $Ca_V1.2$ channels and mCherry-Sec61β in fixed tsA-201 cells were generated using a Leica SR GSD 3D system. The system is built around a Leica DMI6000 B TIRF microscope and is equipped with a Leica oil-immersion HC PL APO 160×/1.43 NA super-resolution objective, four laser lines (405 nm/30 mW, 488 nm/300 mW, 532 nm/500 mW, and 642 nm/500 mW), and an Andor iXon3 EM-CCD. Images were collected in TIRF mode at a frame rate of 100 Hz for 20,000–100,000 frames using Leica Application Suite (LAS AF) software. $Ca_V1.2$ cluster area sizes were determined using binary masks of the images in ImageJ/Fiji.

## Stepwise photobleaching

tsA-201 cells expressing $Ca_V1.2$ channels tagged at their C-terminus with monomeric GFP were fixed in 4% paraformaldehyde (10 min) and imaged in TIRF mode on the Leica 3D-GSD system described above using a 160×/1.43 NA objective. The core Leica DMI6000 B TIRF microscope in this system is capable of functioning outside of GSD SR imaging mode as a conventional TIRF microscope. Cells were illuminated with 488-nm laser light, and image stacks of 2000 frames were acquired at 30 Hz. The first five images after the shutter was opened were averaged, and a rolling-ball background subtraction was applied using ImageJ/Fiji. This image was then low-pass filtered with a 2-pixel cut-off and high-pass filtered with a 5-pixel cut-off (see *Figures 2D and 3D*). Thresholding was then applied to identify connected regions of pixels that were above threshold. The ImageJ/Fiji plugin 'Time Series Analyzer v2.0' was then used to select 4 × 4 pixel regions of interest (ROIs) centered on the peak pixel in each spot. Next, z-axis intensity profiles (where z is time) from these ROIs were examined over the entire image stack. To facilitate the identification of bleaching steps, the signal-to-noise ratio was improved by applying a 5-pixel rolling-ball background subtraction, a median filter (1 pixel radius), and a 10 frame moving average. Bleaching steps were then manually counted.

Identical procedures were performed on cardiomyocytes expressing photo-activatable, GFP-tagged $Ca_V\beta_2$ auxiliary subunits. This subunit binds to the pore-forming $\alpha_1$ subunit of the channel with a 1:1 stoichiometry; thus, expression of this protein in cardiomyocytes represents a photo-activatable fluorescent marker of $Ca_V1.2$ channels. Since adult cardiomyocytes are impervious to chemical transfection, we used adeno-associated virus serotype 9 (AAV9) to transfer this gene into mice via retro-orbital injection, a strategy that has been successfully used by others to transfer cardiac genes in mice (*Fang et al., 2012*). The AAV9-packaged $Ca_V\beta_2$-PA-GFP was engineered from $Ca_V\beta_2$-PA-GFP pcDNA by Vector Biolabs. Mice were sacrificed 5 week after retro-orbital injection. Successful gene transfer was confirmed by photo-activating the $Ca_V\beta_2$-PA-GFP with 405 nm laser light. Prior to photo-activation, no GFP fluorescence emission was detected upon excitation with 488 nm laser light, but after photo-activation, robust GFP fluorescence emission was observed in the z-lines of isolated ventricular myocytes (data not shown). For stepwise photobleaching experiments, GFP was photo-activated prior to starting movie recordings.

## Bimolecular fluorescence complementation

Spontaneous interactions of $Ca_V1.2$ channels were assayed using bimolecular fluorescence complementation. In these experiments, $Ca_V1.2$ channels were tagged at their C-terminus with non-fluorescent N- ($VN_{(1-154, I152L)}$) or C-terminal ($VC_{(155-238, A206K)}$) halves of a 'split' Venus fluorescent protein. When $Ca_V1.2$-VN and $Ca_V1.2$-VC are brought close enough together to interact, the full Venus protein can fold into its functional, fluorescent confirmation. The magnitude of Venus fluorescence emission therefore provides an indicator of $Ca_V1.2$ interactions. Venus fluorescence was monitored in tsA-201 cells expressing $Ca_V1.2$-VN and $Ca_V1.2$-VC using TIRF microscopy, as described above ('recording of $Ca^{2+}$ sparklets'). Transfected cells were identified by co-expression of tagRFP or, in experiments requiring $Ca^{2+}$ imaging with Rhod-2, by weak initial Venus expression.

The relationship between membrane voltage and $Ca_V1.2$ interactions was obtained by subjecting patch-clamped cells in whole-cell mode to a series of 9-s depolarizations from a holding potential of −80 mV to test potentials ranging from −60 to +80 mV. Maturation of newly reconstituted Venus protein takes some time, hence the long depolarizing pulse (*Nagai et al., 2002*). Cells were superfused throughout with the 20 mM $Ca^{2+}$ external solution described above. A TTL pulse generated by the TILL imaging system was used to trigger the onset of each voltage sweep. The cell TIRF footprint was illuminated using 491-nm light throughout each voltage sweep, and PM-localized Venus fluorescence emission was monitored in a TIRF movie acquired at a rate of 100 Hz. The final six frames of each movie were averaged to generate a single image for each voltage sweep. These images were median filtered (1 pixel radius) then divided by the −60 mV image to obtain Venus $F/F_0$ images. Finally, the images were smoothed and pseudo-colored using the 'red hot' lookup table in ImageJ/Fiji. The $Ca^{2+}$ dependence of $Ca_V1.2$ interactions was tested by performing the aforementioned procedure first with 2 mM $Ba^{2+}$ as the conducting ion, then switching the perfusate to our regular external solution described above (20 mM $Ca^{2+}$) and running the procedure again on the same cell (see *Figure 4—figure supplement 2*).

Reconstitution of the Venus protein is irreversible; thus, once channels spontaneously interact, they remain fused together. We exploited this feature of the bimolecular fluorescence complementation assay in experiments designed to investigate the physiological effects of $Ca_V1.2$ interactions. In these experiments, transfected tsA-201 cells expressing the split-Venus–tagged channels were patch-clamped in whole-cell mode and dialyzed with 200 µM Rhod-2 via the pipette. $Ca^{2+}$ sparklets were recorded while holding the cell at −80 mV. The depolarizing step protocol described above was performed, and Venus fluorescence was monitored as before. Increases in Venus fluorescence emission during the protocol provide an indication of $Ca_V1.2$ interactions. Holding cells once more at −80 mV, $Ca^{2+}$ sparklet activity was recorded from the irreversibly fused $Ca_V1.2$ channels.

## Dynamic FRET measurements in live cells

For this set of experiments, tsA-201 cells were transfected with expression constructs for $Ca_V1.2$-EGFP and $Ca_V1.2$-tagRFP. FRET from $Ca_V1.2$-EGFP (donor) to $Ca_V1.2$-tagRFP (acceptor) was measured on an Olympus Fluoview 1000 (FV1000) confocal laser-scanning microscope equipped with an Olympus APON 60×/1.49 NA oil-immersion objective. A 473-nm diode laser was used to excite the sample. Emitted light was separated with an SDM560 beam splitter, collected with BA490-540 and BA575-675 emission filters, and detected by a photomultiplier tube. The resultant raw donor and acceptor images were corrected for background and bleed-through of GFP emission into the RFP channel. In separate experiments using cells expressing only $Ca_V1.2$-tagRFP, bleed-through was determined to be ~16%. Intensity measurements from the corrected images, referred to as GFPg and RFPg respectively (where g refers to excitation of GFP with 473 nm light), were extracted using Metamorph or ImageJ software. FRET was expressed as the ratio, FRETr = RFPg/GFPg. In some experiments, the time course of FRETr was monitored during photolysis of caged $Ca^{2+}$. In others, the time course of FRETr was recorded simultaneously with whole-cell $Ca_V1.2$ currents. A TTL pulse generated by the FV1000 system at the onset of imaging was used to trigger a voltage protocol consisting of a 500-ms depolarizing pulse from a holding potential of −80 to +10 mV. Finally, the $Ca^{2+}$ dependence of FRET between $Ca_V1.2$ channels was monitored while perfusing cells with external solutions containing 5 mM CsCl, 10 mM HEPES, 10 mM glucose, 140 mM NMDG, 1 mM $MgCl_2$ and increasing concentrations (0, 25, 50, 100, 200, 300, 400, 800 and 5000 nM) of $CaCl_2$. The solutions were adjusted to pH 7.4, and 1 µM ionomycin was added to equilibrate intracellular and extracellular $Ca^{2+}$.

## Photolysis of caged $Ca^{2+}$

tsA cells were perfused with a 'zero calcium' external solution containing 5 mM CsCl, 10 mM HEPES, 10 mM glucose, 140 mM NMDG, 1 mM $MgCl_2$, and 2 mM $BaCl_2$ (pH. 7.4). Cells were patch-clamped in whole-cell mode as previously described (*Tadross et al., 2013*) using an internal solution containing 5 mM CsCl, 40 mM HEPES, 135 mM $CsMeSO_3$, 1 mM citrate and 1.6 mM $CaCl_2$, adjusted to pH 7.4 and frozen into aliquots. On the day of experiments, the $Ca^{2+}$ cage DMNP-EDTA (2 mM; Invitrogen, D6814) was added. $Mg^{2+}$ was omitted from the pipette solution since DMNP-EDTA can also cage $Mg^{2+}$. Internal solutions were supplemented with 0.25 µM $PI(4,5)P_2$ (Avanti Polar Lipids, Alabaster, AL; 840046P) and 0.5 µM okadaic acid (LC Laboratories, Woburn, MA; O-2220) to stabilize $I_{Ca}$ (*Tadross et al., 2013*). This enabled us to record stable currents and achieve successful uncaging of $Ca^{2+}$, as assayed with the $Ca^{2+}$-sensitive dyes, Fluo-4 AM or Rhod-2 AM (see *Figure 6A* and *Figure 4—figure supplement 1*). Uncaging experiments were performed on an Olympus FV1000 confocal microscope equipped with a SIM scanner unit that permits simultaneous laser light uncaging (100% 405 nm for two frames) and recording of Venus, Fluo-4, Rhod-2 or EGFP/tagRFP FRET fluorescence emission.

## CaMKII activity assay

CaMKII activity was measured using the 'SignaTECT Calcium/Calmodulin-Dependent Protein Kinase Assay System' (Promega Corporation, Madison, WI). Mouse hearts were homogenized in a lysis buffer solution containing 20 mM Tris–HCl (pH 8.0), 2 mM EDTA, 2 mM EGTA, 2 mM DTT, PhosSTOP phosphatase inhibitor cocktail (Roche Diagnostics GmbH, Mannheim, Germany) and cOmplete, Mini protease inhibitor cocktail (Roche Diagnostics). Hearts were subjected to three 5 s pulses at 12,000–17,000 rpm using a PRO200 Homogenizer Unit (Pro Scientific, Oxford, CT). The homogenate was centrifuged at 2000 rpm for 10 min and the resultant supernatant was collected and assayed for CaMKII activity as per the manufacturers instructions. To determine the effect of MLCKp on CaMKII activity, 0.1 or 1 µM MLCKp was added to the reaction.

## Acknowledgements

We thank Dr Joshua Vaughan for help with super-resolution imaging procedures and analyses. Drs William Catterall and Ruth E Westenbroek provided antibodies against $Ca_V1.2$. The work was supported by grants from the NIH (HL085870, HL085686 and NS077863) and AHA (14GRNT18730054).

## Additional information

### Funding

| Funder | Grant reference number | Author |
| --- | --- | --- |
| National Institutes of Health (NIH) | HL085870 | Luis F Santana |
| American Heart Association (AHA) | 14GRNT18730054 | Manuel F Navedo |
| National Institutes of Health (NIH) | HL085686 | Luis F Santana |
| National Institutes of Health (NIH) | NS077863 | Marc D Binder |

The funders had no role in study design, data collection and interpretation, or the decision to submit the work for publication.

### Author contributions

RED, Conception and design, Acquisition of data, Analysis and interpretation of data, Drafting or revising the article; CMM, Acquisition of data, Analysis and interpretation of data, Drafting or revising the article; CY, XO-A, Acquisition of data, Analysis and interpretation of data, Contributed unpublished essential data or reagents; MDB, MFN, Conception and design, Drafting or revising the article; LFS, Conception and design, Analysis and interpretation of data, Drafting or revising the article

### Ethics

Animal experimentation: This study was performed in strict accordance with the recommendations in the Guide for the Care and Use of Laboratory Animals of the National Institutes of Health. All of the animals were handled according to approved institutional animal care and use committee (IACUC) protocols (#3374-01) of the University of Washington.

## Additional files

### Supplementary files

- Source code 1. Custom software for $Ca^{2+}$ sparklet detection and analysis written in MATLAB.

- Source code 2. Binary coupled Markov chain model.

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
