## [Decision Letter]

Thank you for sending your work entitled “Graded Ca^2+/^calmodulin-dependent coupling of voltage-gated Ca_V_1.2 channels” for consideration at *eLife*. Your article has been favorably evaluated by Eve Marder (Senior editor) and 3 reviewers, one of whom is a member of our Board of Reviewing Editors.

The following individuals responsible for the peer review of your submission have agreed to reveal their identity: Richard Aldrich (Reviewing editor) and Robert Kass (peer reviewer). A further reviewer remains anonymous.

The Reviewing editor and the other reviewers discussed their comments before we reached this decision, and the Reviewing editor has assembled the following comments to help you prepare a revised submission.

The present work is a follow up of a previous study by the same group published in 2012 (Dixon et al., PNAS) and in which they showed that Ca_V_1.2 channels form clusters in ventricular myocytes and that oligomerisation of L-type calcium channels via their C-termini result in the amplification of Ca^2+^ influx into cells. In the present work, the authors use a combination of super resolution nanoscopy, Ca^2+^ imaging, electrophysiology and two methodologically distinct assays of protein-protein interaction to investigate the molecular basis for Ca_V_1.2 channels oligomerisation and its functional impact. The goal of this work is to determine the mechanisms allowing the simultaneous opening of calcium channels near the junctional SR in order to explain the high probability of RYR activation during the cardiac AP. The main findings of this study are that: i) Ca_v_1.2 channels form clusters containing 5 channels in tsA201 cells and 8 channels in mouse ventricular myocytes; ii) within these clusters, Ca_v_1.2 channels interact with each other via their C-termini and this interaction, which can occur in presence of elevated cytosolic Ca^2+^, requires Ca^2+^ entry through L-type calcium channels; iii) CaM molecules are involved in Ca_v_1.2-to-Ca_v_1.2 interaction via their C-termini by binding to a pre-IQ sequence and this is also required for Ca_v_1.2 channels clustering; iv) the functional impact of Ca_v_1.2 channel coupling within a cluster consists in the instantaneous opening of channels together (and subsequent closing together) which is associated with multi-quantal Ca^2+^ sparklets. Also, Ca_v_1.2 channels coupling transiently persists after L-type calcium current has entered the cell and this is associated with Ca^2+^-dependent facilitation in cardiac myocytes. In addition, the possible dynamic clustering and de-clustering of Ca_v_1.2 channels provides a mechanism of molecular memory, in which, the temporal limitation of channel disassociation following a first event could facilitation the activation of the subsequent events. These data provide novel insights of intracellular signaling networks of facilitation and are reminiscent of long-term potentiation and synaptic plasticity in neuronal networks.

This is an excellent paper that has no substantial flaws. However there are a few relatively minor issues that need to be addressed.

Specific comments:

1) Using high resolution nanoscopy, the authors elegantly show that Ca_v_1.2 channels localize along the t-tubules and SR junctions of isolated cardiac myocytes. If these small SR microdomains are important for the localized signaling (i.e., channel clustering) and lead to Ca_v_1.2 activation and facilitation, wouldn't the authors predict that in their tsA-201 expression, the Ca_v_1.2 clusters would co-localize with SR/ER structures? This should be discussed.

2) The authors bring up an interesting concept of “molecular memory”, in which following the termination of the first event, Ca_v_1.2 de-clustering persist and this lag time could transiently facilitate an increase in the Ca^2+^-sensitivity and activation of subsequent events. This “priming” through maintenance of clusters (or slow dissociated of channels from clusters) would therefore be dependent on magnitude and/or time of the first, or cluster-inducing stimulus. In the experiments performed in Figure 7, does the increase in V2 amplitude (in the present of Ca^2+^) depend on the length of the V_pre_ pulse? Is there a temporal threshold needed to elicit a second facilitated response? And could the mechanism of clustering and de-clustering of Ca_v_1.2 be a possible Ca^2+^ sensitive graded response.

---

## [Author Response]

*1) Using high resolution nanoscopy, the authors elegantly show that Ca*_*v*_*1.2 channels localize along the t-tubules and SR junctions of isolated cardiac myocytes. If these small SR microdomains are important for the localized signaling (i.e., channel clustering) and lead to Ca*_*v*_*1.2 activation and facilitation, wouldn't the authors predict that in their tsA-201 expression, the Ca*_*v*_*1.2 clusters would co-localize with SR/ER structures? This should be discussed*.

The reviewers pose an interesting question about potential co-localization of Ca_v_1.2 channel clusters and the ER in tsA-201 cells. Epithelial tsA-201 cells (derived from HEK293 cells) are neither excitable nor contractile and lack a specialized ryanodine receptor-mediated Ca^2+^ release mechanism. Accordingly the tsA-201 cells do not possess the same specialized architecture or couplon structures observed in cardiac myocytes. Further, these cells do not express detectable levels of endogenous Ca_v_1.2 channels (Perez-Garcia et al., 1995) or ryanodine receptors (Rossi et al., 2002; Tong et al., 1999). Nonetheless, we concede the possibility that heterologous expression of Ca_v_1.2 channels in these cells might result in their co-localization with ER structures that are present in the cells. To address this question, we used super-resolution imaging to examine Ca_v_1.2 channel-expressing tsA-201 cells double labeled with antibodies directed against: (i) Ca_v_1.2 channels and, (ii) mCherry-Sec61β (a general ER marker; [56]). The results of these experiments, presented in Figure 3—figure supplement 1, revealed that Ca_v_1.2 channel clusters are not preferentially co-localized to tsA-201 ER structures. We further explored this idea by co-transfecting tsA-201 cells with the membrane-binding protein Junctophilin-2 (JPH2). This protein is known to stabilize the junctional membrane complex in cardiomyocytes by tethering the junctional SR membrane to the t-tubule membrane (45). We used JPH2-GFP to increase anchor points between the ER and PM in tsA-201 cells. However, even in the presence of JPH2, Ca_v_1.2 channel clusters were not distributed along the PM-ER junctions in the manner that they are in cardiomyocytes. These data, now presented in Figure 3—figure supplement 1, suggest that Ca_v_1.2 channel clustering occurs independently of SR/ER microdomains.

We performed additional imaging experiments on tsA-201 cells expressing Ca_v_1.2 channels with the ‘pre-IQ swap’ mutation that prevents Ca_v_1.2 channel interactions (assayed with BiFC and presented in Figure 4—figure supplement 4). In these experiments, we found that Ca_v_1.2(pre-IQ swap) channels still formed clusters in tsA-201 cells, despite their inability to interact functionally. Indeed, under identical imaging conditions (i.e., using the same fixative and TIRF penetration depth), Ca_v_1.2 (pre-IQ swap) channel cluster areas were not significantly different from those of WT channels. These results suggest that while the physical proximity of Ca_v_1.2 channels is necessary for channel interactions, it is not sufficient for their functional coupling. These data have now been added to Figure 4—figure supplement 5. The implications of these new data are that the clustered arrangement of Ca_v_1.2 channels may be dictated by another unknown protein scaffold. The identity of these proteins will be sought in future studies.

*2) The authors bring up an interesting concept of “molecular memory”, in which following the termination of the first event, Ca*_*v*_*1.2 de-clustering persist and this lag time could transiently facilitate an increase in the Ca*^*2+*^*-sensitivity and activation of subsequent events. This “priming” through maintenance of clusters (or slow dissociated of channels from clusters) would therefore be dependent on magnitude and/or time of the first, or cluster-inducing stimulus. In the experiments performed in*
Figure 7*, does the increase in V2 amplitude (in the present of Ca*^*2+*^*) depend on the length of the V*_*pre*_
*pulse? Is there a temporal threshold needed to elicit a second facilitated response? And could the mechanism of clustering and de-clustering of Ca*_*v*_*1.2 be a possible Ca*^*2+*^
*sensitive graded response*.

We thank the reviewers for raising these interesting points. We want to clarify that we are not suggesting that channels dynamically cluster and de-cluster but rather that channels within pre-defined clusters dynamically and functionally interact with one another during calcium influx events. This is how we considered the reviewers question in our response. We now include additional experiments designed to address these questions. Specifically, we measured I_2_/I_1_ current amplitude ratios with interpulse intervals ranging from 0.05 to 1.6 s in control ventricular myocytes and in myocytes dialyzed with 100 nM or 1 μM MLCKp. In control myocytes, peak levels of I_Ca_ facilitation were apparent with an interpulse interval of 300 ms and declined thereafter whereas facilitation was absent at all interpulse-interval durations in myocytes superfused with Ba^2+^ or dialyzed with 1 μM MLCKp. These results are consistent with our hypothesis that Ca^2+^ ions, not Ba^2+^ ions, augment Ca_V_1.2 channel activity, likely by increasing the number of functionally coupled channels. Furthermore, these experiments reveal the “priming” phenomenon that we had proposed based on our FRET experiments. These new and exciting data have now been added to Figure 7.

*References*:

Perez-Garcia, M.T., Kamp, T.J., and Marban, E. (1995). Functional properties of cardiac L-type calcium channels transiently expressed in HEK293 cells. Roles of alpha 1 and beta subunits. J Gen Physiol 105, 289-305.10.1085/jgp.105.2.289

Rossi, D., Simeoni, I., Micheli, M., Bootman, M., Lipp, P., Allen, P.D., and Sorrentino, V. (2002). RyR1 and RyR3 isoforms provide distinct intracellular Ca^2+^ signals in HEK 293 cells. J Cell Sci 115, 2497-2504

Takeshima, H., Komazaki, S., Nishi, M., Iino, M., and Kangawa, K. (2000). Junctophilins: a novel family of junctional membrane complex proteins. Molecular cell 6, 11-22.10.1016/S1097-2765(05)00005-5

Tong, J., Du, G.G., Chen, S.R., and MacLennan, D.H. (1999). HEK-293 cells possess a carbachol- and thapsigargin-sensitive intracellular Ca^2+^ store that is responsive to stop-flow medium changes and insensitive to caffeine and ryanodine. The Biochemical Journal 343 Pt 1, 39-44. 0.1042/0264-6021:3430039

Zurek, N., Sparks, L., and Voeltz, G. (2011). Reticulon short hairpin transmembrane domains are used to shape ER tubules. Traffic 12, 28-41.

10.1111/j.1600-0854.2010.01134.x